# An Inductive Bias for Distances: Neural Nets that Respect the Triangle Inequality

**Silviu Pitis\*, Harris Chan\*, Kiarash Jamali, Jimmy Ba**
University of Toronto, Vector Institute
`{spitis, hchan}@cs.toronto.edu`

## Abstract

Distances are pervasive in machine learning. They serve as similarity measures, loss functions, and learning targets; it is said that a good distance measure solves a task. When defining distances, the triangle inequality has proven to be a useful constraint, both theoretically—to prove convergence and optimality guarantees—and empirically—as an inductive bias. Deep metric learning architectures that respect the triangle inequality rely, almost exclusively, on Euclidean distance in the latent space. Though effective, this fails to model two broad classes of subadditive distances, common in graphs and reinforcement learning: asymmetric metrics, and metrics that cannot be embedded into Euclidean space. To address these problems, we introduce novel architectures that are guaranteed to satisfy the triangle inequality. We prove our architectures universally approximate norm-induced metrics on $\mathbb{R}^n$, and present a similar result for modified Input Convex Neural Networks. We show that our architectures outperform existing metric approaches when modeling graph distances and have a better inductive bias than non-metric approaches when training data is limited in the multi-goal reinforcement learning setting.[1]

## 1 Introduction

Many machine learning tasks involve a distance measure over the input domain. A good measure can make a once hard task easy, even trivial. In many cases—including graph distances, certain clustering algorithms, and general value functions in reinforcement learning (RL)—it is either known that distances satisfy the triangle inequality, or required for purposes of theoretical guarantees; e.g., speed and loss guarantees in $k$-nearest neighbors and clustering (Cover and Hart, 1967; Indyk, 1999; Davidson and Ravi, 2009), or optimality guarantees for $A^*$ search (Russell and Norvig, 2016). This also makes the triangle inequality a potentially useful *inductive bias* for learning distances. For these reasons, numerous papers have studied different ways to learn distances that satisfy the triangle inequality (Xing et al., 2003; Yang and Jin, 2006; Brickell et al., 2008; Kulis et al., 2013).

The usual approach to enforcing the triangle inequality in deep metric learning (Yi et al., 2014; Hoffer and Ailon, 2015; Wang et al., 2018) is to use a Siamese network (Bromley et al., 1994) that computes a Euclidean distance in the latent space. Specifically, the Siamese network models distance $d_{\mathcal{X}} : \mathcal{X} \times \mathcal{X} \to \mathbb{R}^+$ on domain $\mathcal{X}$ by learning embedding $\phi : \mathcal{X} \to \mathbb{R}^n$ and computing $d_{\mathcal{X}}(x, y)$ as $\|\phi(x) - \phi(y)\|_2$. Successful applications include collaborative filtering (Hsieh et al., 2017), few-shot learning (Snell et al., 2017), and multi-goal reinforcement learning (Schaul et al., 2015). The use of Euclidean distance, however, has at least two downsides. First, the Euclidean architecture cannot represent asymmetric metrics, which arise naturally in directed graphs and reinforcement learning. Second, it is well known that for some metric spaces $(\mathcal{X}, d_{\mathcal{X}})$, including large classes of symmetric graphs (e.g., constant-degree expanders and $k$-regular graphs), there is *no* embedding $\phi : \mathcal{X} \to \mathbb{R}^n$ that can model $d_{\mathcal{X}}$ precisely using $\|\cdot\|_2$ (Indyk et al., 2017). A classic example is shown in Figure 1.

In part due to these issues, some have considered non-architectural constraints. He et al. (2016) impose a triangle inequality constraint in RL via an online, algorithmic penalty. Implementing such a penalty can be expensive, and does not provide any guarantees. An approach that does guarantee

---

[1]Code available at `https://github.com/spitis/deepnorms`

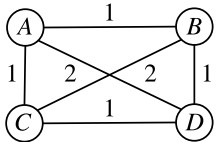

| Norm | MSE | Deep Norm | Wide Norm | Mahalanobis |
|---|---|---|---|---|
| Euclidean, $\mathbb{R}^n, \forall n$ | 0.057 | | | |
| Deep Norm, $\mathbb{R}^2$ | 0.000 | | | |
| Wide Norm, $\mathbb{R}^2$ | 0.000 | | | |

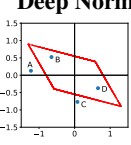 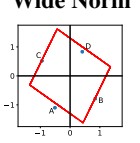 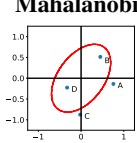

**Fig. 1:** The nodes in the graph (left) cannot be embedded into *any* $\mathbb{R}^n$ so that edge distances are represented by the Euclidean metric: points $\phi(A)$ and $\phi(D)$ must lie at the midpoint of the segment from $\phi(B)$ to $\phi(C)$—but then $\phi(A)$ and $\phi(D)$ coincide, which is incorrect. Our models fit the data in $\mathbb{R}^2$ (middle). The visualization (right) shows learned norm balls in red and embeddings in blue.

satisfaction of triangle inequality is to fix any violations after learning, as done by Brickell et al. (2008). But this does not scale to large problems or provide an inductive bias during learning.

Is it possible to impose the triangle inequality architecturally, without the downsides of Euclidean distance?

In response to this question, we present the following contributions: (1) three novel neural network architectures, Deep Norms, Wide Norms and Neural Metrics, which model symmetric and asymmetric norms and metrics, (2) universal approximation theorems for Deep Norms and Wide Norms and modified Input Convex Neural Networks (Amos et al., 2017), and (3) empirical evaluations of our models on several tasks: modeling norms, metric nearness, modeling shortest path lengths, and learning a general value function (Sutton et al., 2011). Our models are guaranteed to satisfy the triangle inequality, straightforward to implement, and may be used in place of the usual Euclidean metric should one seek to model asymmetry or increase expressiveness.

## 2 MODELING NORMS

### 2.1 PRELIMINARIES

Our goal is to construct expressive models of metrics and quasi-metrics on domain $\mathcal{X}$. A **metric** is a function $d : \mathcal{X} \times \mathcal{X} \to \mathbb{R}^+$ satisfying, $\forall x, y, z \in \mathcal{X}$:

**M1** (Non-negativity)**.** $d(x, y) \geq 0$.     **M3** (Subadditivity)**.** $d(x, z) \leq d(x, y) + d(y, z)$.

**M2** (Definiteness)**.** $d(x, y) = 0 \iff x = y$.     **M4** (Symmetry)**.** $d(x, y) = d(y, x)$.

Since we care mostly about the triangle inequality (**M3**), we relax other axioms and define a **quasi-metric** as a function that is **M1** and **M3**, but not necessarily **M2** or **M4**. Given weighted graph $G = (\mathcal{V}, \mathcal{E})$ with non-negative weights, shortest path lengths define a quasi-metric between vertices.

When $\mathcal{X}$ is a vector space (we assume over $\mathbb{R}$), many common metrics, e.g., Euclidean and Manhattan distances, are induced by a norm. A **norm** is a function $\| \cdot \| : \mathcal{X} \to \mathbb{R}$ satisfying, $\forall x, y \in \mathcal{X}, \alpha \in \mathbb{R}^+$:

**N1** (Pos. def.)**.** $\|x\| > 0$, *unless* $x = 0$.     **N3** (Subadditivity)**.** $\|x + y\| \leq \|x\| + \|y\|$.

**N2** (Pos. homo.)**.** $\alpha\|x\| = \|\alpha x\|$, *for* $\alpha \geq 0$.     **N4** (Symmetry)**.** $\|x\| = \|-x\|$.

An **asymmetric norm** is **N1**-**N3**, but not necessarily **N4**. An **(asymmetric) semi-norm** is non-negative, **N2** and **N3** (and **N4**), but not necessarily **N1**. We will use the fact that any asymmetric semi-norm $\| \cdot \|$ induces a quasi-metric using the rule, $d(x, y) = \|x - y\|$, and first construct models of asymmetric semi-norms. Any induced quasi-metric $d$ is translation invariant—$d(x, y) = d(x + z, y + z)$—and positive homogeneous—$d(\alpha x, \alpha y) = \alpha d(x, y)$ for $\alpha \geq 0$. If $\| \cdot \|$ is symmetric (**N4**), so is $d$ (**M4**). If $\| \cdot \|$ is **N1**, $d$ is **M2**. Metrics that are not translation invariant (e.g., Bi et al. (2015)) or positive homogeneous (e.g., our Neural Metrics in Section 3) cannot be induced by a norm.

A convex function $f : \mathcal{X} \to \mathbb{R}$ is a function satisfying **C1:** $\forall x, y \in \mathcal{X}, \alpha \in [0, 1]$: $f(\alpha x + (1 - \alpha)y) \leq \alpha f(x) + (1 - \alpha)f(y)$. The commonly used ReLU activation, $\mathrm{relu}(x) = \max(0, x)$, is convex.

### 2.2 DEEP NORMS

It is easy to see that any **N2** and **N3** function is convex—thus, all asymmetric semi-norms are convex. This motivates modeling norms as constrained convex functions, using the following proposition.

**Proposition 1.** *All positive homogeneous convex functions are subadditive; i.e.,* ***C1*** $\wedge$ ***N2*** $\Rightarrow$ ***N3***.

The proof is straightforward (put $\alpha = \frac{1}{2}$ in **C1** and apply **N2** to the left side). To use Proposition 1, we begin with the Input Convex Neural Network (ICNN) (Amos et al., 2017) architecture, which satisfies **C1**, and further constrain it to be non-negative and satisfy **N2**. The resulting **Deep Norm** architecture is guaranteed to be an asymmetric semi-norm. A $k$-layer Deep Norm is defined as:

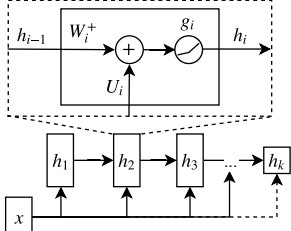

$$\|x\| = h_k, \qquad \text{with} \qquad h_i = g_i(W_i^+ h_{i-1} + U_i x) \qquad (1)$$

**Fig. 2:** Deep Norm architecture.

for $i = 1 \ldots k$, where $x$ is the input, $h_0 = 0, W_1^+ = 0$, the activation functions $g_i$ preserve **C1** and **N2** (element-wise), $g_k$ is non-negative, $W_i^+$ is a non-negative matrix, and $U_i$ is an unconstrained matrix.

As compared to the original ICNN architecture, we have omitted the bias terms from Equation 1, have constrained the $g_i$ to preserve positive homogeneity while also allowing them to be *any* function that preserves element-wise convexity (this is essential to our universal approximation results), and have required $g_k$ to be non-negative. It is easy to verify that the set of valid *element-wise $g_i$* is $\{g_{\alpha\beta}(x) = \alpha\,\mathrm{relu}(x) + \beta x\,|\,\alpha, \beta \geq 0\}$. This includes ReLUs and leaky ReLUs. But we do not restrict ourselves to element-wise activations. Inspired by GroupSort (Anil et al., 2018), we use activations that depend on multiple inputs (and preserve element-wise **C1** and **N2**). In particular, we use the pairwise *MaxReLU*:

$$\mathrm{maxrelu}(x, y) = [\max(x, y), \alpha\,\mathrm{relu}(x) + \beta\,\mathrm{relu}(y)], \text{ where } \alpha, \beta \geq 0 \qquad (2)$$

Deep Norms are **N2** and **N3**. Using the following propositions, we may also impose **N1** and **N4**.

**Proposition 2.** *If $\|\cdot\|$ is an asymmetric semi-norm, then $\|x\| = \|x\| + \|-x\|$ is a semi-norm.*

**Proposition 3.** *If $\|\cdot\|_a$ is an (asymmetric) semi-norm, $\|\cdot\|_b$ is a norm (e.g., $\|\cdot\|_b = \|\cdot\|_2$), and $\lambda > 0$, then $\|x\|_{a+\lambda b} = \|x\|_a + \lambda\|x\|_b$ is an (asymmetric) norm.*

## 2.3 WIDE NORMS

In addition to Deep Norms, we propose the following alternative method for constructing norms: a Wide Norm is any combination of (asymmetric) (semi-) norms that preserves **N1-N4**. It is easy to verify that both (1) non-negative sums and (2) $\max$ are valid combinations (indeed, these properties were also used to construct Deep Norms), and so the vector-wise *MaxMean* combination is valid: $\mathrm{maxmean}(x_1, x_2, \ldots, x_n) = \alpha\,\max(x_1, x_2, \ldots, x_n) + (1 - \alpha)\,\mathrm{mean}(x_1, x_2, \ldots, x_n)$.

Although the family of Wide Norms is broad, for computational reasons to be discussed in Subsection 3.5, we focus our attention on the Wide Mahalanobis norm. References to "Wide Norms" in the rest of this paper refer to Wide Mahalanobis norms. The **Mahalanobis** norm of $x \in \mathbb{R}^n$, parameterized by $W \in \mathbb{R}^{m \times n}$, is defined as $\|x\|_W = \|Wx\|_2$. It is easily verified that $\|\cdot\|_W$ is a proper norm when $W$ is a non-singular (square) matrix, and a semi-norm when $W$ is singular or $m < n$.

A $k$-component Mixture of Mahalanobis norm (hereafter **Wide Norm**, or Wide Norm with $k$ Euclidean components) is defined as the $\mathrm{maxmean}$ of $k$ Mahalanobis norms:

$$\|x\| = \mathrm{maxmean}_i \left(\|W_i x\|_2\right) \qquad \text{where} \quad W_i \in \mathbb{R}^{m_i \times n} \text{ with } m_i \leq n. \qquad (3)$$

Wide Norms are symmetric by default, and must be *asymmetrized* to obtain asymmetric (semi-) norms. We use the below property (Bauer et al., 1961) and propositions (proofs in Appendix A).

**N5.** *$\|\cdot\|$ is **monotonic in the positive orthant** if $0 \leq x \leq y$ (element-wise) implies $\|x\| \leq \|y\|$.*

**Proposition 4.** *If $\|\cdot\|$ is an N5 (semi-) norm on $\mathbb{R}^{2n}$, then $\|x\| = \|\mathrm{relu}(x :: -x)\|$, where $::$ denotes concatenation, is an asymmetric (semi-) norm on $\mathbb{R}^n$.*

**Proposition 5.** *The Mahalanobis norm with $W = DU$, with $D$ diagonal and $U$ non-negative, is N5.*

## 2.4 UNIVERSAL APPROXIMATION OF CONVEX FUNCTIONS AND NORMS

How expressive are Deep Norms and Wide Norms? Although it was empirically shown that ICNNs have "substantial representation power" (Amos et al., 2017), the only prior work characterizing the approximation power of ICNNs uses a narrow network with infinite depth, which does not reflect typical usage (Chen et al., 2018). One of our key contributions is a series of universal approximation results for ICNNs (with MaxReLU activations), Deep Norms and Wide Norms that use a more practical infinite width construction. The next lemma is central to our results.

**Lemma 1** (Semilattice Stone-Weierstrass (from below)). *Let $C$ be a set of continuous functions defined on compact subset $K$ of $\mathbb{R}^n$, $L$ be a closed subset of $C$, and $f \in C$. If (1) for every $x \in K$, there exists $g_x \in L$ such that $g_x \leq f$ and $g_x(x) = f(x)$, and (2) $L$ is closed under $\max$ (i.e., $a, b \in L \Rightarrow \max(a, b) \in L$), then $\exists h \in L$ with $f = h$ on $K$.*

Intuitively, Lemma 1 and its proof (Appendix A) say we can approximate continuous $f$ arbitrarily well with a family $L$ of functions that is closed under maximums if we can "wrap" $f$ from below using functions $g_x \in L$ with $g_x \leq f$. Our Universal Approximation (UA) results are now straightforward.

**Theorem 1** (UA for MICNNs). *The family $\mathcal{M}$ of Max Input Convex Neural Networks (MICNNs) that uses pairwise max-pooling (or MaxReLU) activations is dense in the family $\mathcal{C}$ of convex functions.*

*Proof.* For $f \in \mathcal{C}$, $x \in \mathbb{R}^n$, let $g_x \in \mathcal{M}$ be a linear function whose hyperplane in the graph of $f$ is tangent to $f$ at $x$. Then $g_x$ satisfies condition (1) of Lemma 1 (because $f$ is convex). The use of pairwise $\max$ activations allows one to construct $\max(h_1, h_2) \in \mathcal{M}$ for any two $h_1, h_2 \in \mathcal{M}$ by using $\log_2(n)$ max-pooling layers, satisfying condition (2) of Lemma 1. Thus $f$ is in the closure of $\mathcal{M}$, and the result follows. □

This result applies when MaxReLUs are used, since we can set $\alpha, \beta = 0$. Using MaxReLU (rather than $\max$) guarantees that MICNNs can imitate regular ICNNs with at most double the parameters.

**Theorem 2** (UA for Deep Norms and Wide Norms). *The families $\mathcal{D}$ of Deep Norms (using MaxReLU) and $\mathcal{W}$ of Wide Norms (using MaxMean) are dense in the family $\mathcal{N}$ of asymmetric semi-norms.*

*Proof (sketch).* The proof is almost identical to that of Theorem 1, except that here $\mathcal{D}$ and $\mathcal{W}$ contain all linear functions whose graph is tangent to any $f \in \mathcal{N}$ since $f$ is **N2**. This is easy to see for functions defined on $\mathbb{R}^2$. See Appendix A for more details. □

## 2.5 Modeling Norms in 2D

Having shown that Deep Norms and Wide Norms universally approximate norms, we now show that they can successfully *learn* to approximate random norms on $\mathbb{R}^2$ when trained using gradient descent. To generate data, we use the below fact, proved in Appendix A.

**Proposition 6.** *The set of all asymmetric norms on $\mathbb{R}^n$ is in one-to-one correspondence with the set of all bounded and open convex sets ("unit balls") containing the origin.*

We use Proposition 6 by generating a random point set, computing its convex hull, and using the hull as the unit ball of the generated norm, $\| \cdot \|$. We then train different models to approximate $\| \cdot \|$ using $|D| \in \{16, 128\}$ training samples of form $((x\eta, y\eta), \eta)$, where $\|(x, y)\| = 1$ and $\eta \sim \mathcal{U}(0.85, 1.15)$. The models are trained to minimize the mean squared error (MSE) between the predicted (scalar) norm value and the ground truth norm value. See Appendix C.1 for details.

Figure 3 illustrates the learned 2D norm balls of a random symmetric (top) and asymmetric (bottom) norm, for three architectures: Deep Norm, Wide Norm, and an unconstrained, fully connected neural network (MLP). The blue contours in Figure 3 are the ground truth norm balls for values $\{0.5, 1, 1.5\}$, and black contours the norm balls of the learned approximations. With the small dataset, the MLP

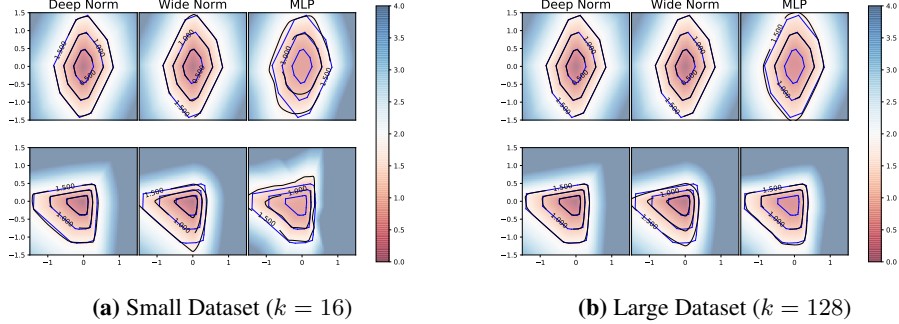

(a) Small Dataset ($k = 16$)        (b) Large Dataset ($k = 128$)

**Fig. 3:** Visualization of learned 2D norms (top) and asymmetric norms (bottom). Blue contours are the ground truth norm balls for values $\{0.5, 1, 1.5\}$, and black contours the learned approximations.

| | N1 (M1-2) | N2 (Homo.) | N3 (M3) | N4 (M4) | UA | Notes |
|---|---|---|---|---|---|---|
| Euclidean | ✓ | ✓ | ✓ | ✓ | ✗ | |
| MLP | ✗ | ✗ | ✗ | ✗ | ✓ | |
| **Deep Norm** | ✶ | ✓ | ✓ | ✶ | ✓ | |
| **Wide Norm** | ✶ | ✓ | ✓ | ✶ | ✓ | works for large minibatches (§§3.5) |
| **Neural Metric** | ✶ | ✶ | ✓ | ✶ | ✓ | based on Deep Norm or Wide Norm |

**Table 1:** Norm (metric) properties of different architectures. As compared to Euclidean architectures, ours are universal asymmetric semi-norm approximators (UA) and can use propositions to optionally satisfy (✶) **N1** and **N4**. Neural metrics relax the unnecessary homogeneity constraint on metrics.

overfits to the training data and generalizes poorly, while Deep Norm and Wide Norm generalize almost perfectly. With the large dataset, we observe that Deep Norms and Wide Norms generalize to larger and smaller norm balls (due to being **N2**), whereas the MLP is unable to generalize to the 0.5 norm ball (indeed, $\text{MLP}^{-1}(0.5) = \emptyset$). While the symmetric Wide Norm fits well, the results suggest that the asymmetrized Wide Norm is not as effective as the naturally asymmetric Deep Norm. See Appendix C for additional details and visualizations.

# 3 MODELING METRICS

Having constructed models of asymmetric semi-norms on $\mathbb{R}^n$, we now use them to induce quasi-metrics on a domain $\mathcal{X}$. As the geometry of $\mathcal{X}$'s raw feature space will not, in general, be amenable to a favorable metric (indeed, the raw features need not be in $\mathbb{R}^n$), we assume the Siamese network approach and learn an embedding function, $\phi : \mathcal{X} \to \mathbb{R}^n$. A Deep Norm or Wide Norm $\|\cdot\|_\theta$, with parameters $\theta$, is defined on $\mathbb{R}^n$ and the metric over $\mathcal{X}$ is induced as $d_{\phi,\theta}(x,y) = \|\phi(y) - \phi(x)\|_\theta$. We could also define $\|\cdot\|_\theta$ using an unconstrained neural network (MLP), but this would not be guaranteed to satisfy the norm axioms and induce a quasi-metric.

To illustrate this approach, we revisit Figure 1. To get the results shown, we represent the four nodes as one hot vectors and embed them into $\mathbb{R}^2$ using $\phi(x) = Wx$, $W \in \mathbb{R}^{2 \times 4}$. We then define the norm on $\mathbb{R}^2$ as either a Mahalanobis norm, a Deep Norm, or a Wide Norm. Training the norm and $\phi$ together, end-to-end with gradient descent, produces the Figure 1 results.

The choices are summarized in Table 1. While Deep Norm and Wide Norm have similar properties, the depth of the former suggests that they can learn efficient, hierarchical representations (as do deep neural networks); however, Wide Norms have a computational advantage when computing pairwise distances for large minibatches, which we explore in Subsection 3.5. Since inducing metrics with Deep Norms or Wide Norms produces a potentially unnecessary homogeneity constraint (due to **N2**), the next Subsection considers Neural metrics, which offer an approach to relaxing this constraint.

## 3.1 NEURAL METRICS

Instead of inducing a metric with a norm, we could define a metric directly as a non-negative weighted sum of different metrics. E.g., as we did for Wide Norms, we can define a **Wide Metric** as the mean of $k$ deep Euclidean metrics. If all components of a Wide Metric are norm-induced, however, it can be induced directly by a single Wide Norm with $k$ components, by setting $\phi(x)$ to be the concatenation of the $\phi_i(x)$, and using each $W_j$ to select the indices corresponding to $\phi_j$. It follows that for a family of Wide Metrics to be more expressive than the family of norm-induced metrics, it must include components that are either not translation invariant or not positive homogeneous. We consider the latter, and use the propositions below to modify our norm-induced metrics (proofs in Appendix A).

**Proposition 7** (Metric-preserving concave functions). *If $d : \mathcal{X} \times \mathcal{X} \to \mathbb{R}^+$ is (quasi-) metric, $f : \mathbb{R}^+ \to \mathbb{R}^+$, $f^{-1}(0) = \{0\}$, and $f$ is concave (i.e., $-f$ is convex), then $f \circ d$ is (quasi-) metric.*

**Proposition 8** (Max preserves metrics). *If $d_1$ and $d_2$ are (quasi-) metric, so too is $\max(d_1, d_2)$.*

To use Proposition 7, we note that each unit in the final layer of a Deep Norm defines a metric (assuming $g_{k-1}$ is non-negative, as it is when using ReLU or MaxReLU activations), as does

each component of a Wide Norm. Thus, by applying metric-preserving functions $f_i$ to these metrics, and then combining them using a MaxMean, mean or max, we obtain a valid metric, which we name a **Neural Metric**. Our $k$-component $f_i$ are parameterized by $w_i, b_i \in \mathbb{R}^k$ as follows: $f_i(x) = \min_j \{w_{ij}x + b_{ij}\}$, where $w_{ij} \geq 0$, $b_{ij} \geq 0$, $b_0 = 0$.

The advantage of Neural Metrics over plain Deep Norms and Wide Norms is that one can better model certain metrics, such as $d(x, y) = \min(1, \|x - y\|)$. This type of metric might be induced by shortest path lengths in a fully connected graph with some maximum edge weight (for example, a navigation problem with a teleportation device that takes some constant time to operate).

## 3.2 APPLICATION: METRIC NEARNESS

We now apply our models to the *metric nearness* problem (Sra et al., 2005; Brickell et al., 2008). In this problem, we are given a matrix of pairwise non-metric distances between (finite) $n$ objects, and it is desired to minimally repair the data to satisfy metric properties (the triangle inequality, **M3**, in particular). Formally, given data matrix $D \in \mathbb{R}^{n \times n}$, we seek metric solution $X \in \mathbb{R}^{n \times n}$ that minimizes the "distortion" $J_{MN} = \|X - D\|_2 / \|D\|_2$ (normalized distance) to the original metric. Sra et al. (2005) proved this loss function attains its unique global

|      | $J_{MN}^{(S)}$ | $\#_{\neg \mathbf{M3}}^{(S)}$ | $J_{MN}^{(A)}$ | $\#_{\neg \mathbf{M3}}^{(A)}$ |
|------|------|------|------|------|
| TF   | 2.01e-2 | 7.7e3 | 1.30e-1 | 3.8e2 |
| Eucl | 9.12e-2 | 0 | 2.02e-1 | 0 |
| WN   | 2.44e-2 | 0 | 1.86e-1 | 0 |
| DN   | **2.00e-2** | 0 | **7.29e-2** | 0 |

**Table 2:** Metric nearness for sym. (S) and asym. (A) matrices in $\mathbb{R}^{200 \times 200}$ (10 seeds).

minimum in the set of size $n$ discrete metrics and proposed an $O(n^3)$ *triangle fixing* ("TF") algorithm for the symmetric case, which iteratively fixes **M3** violations and is guaranteed to converge to the global minimum. Table 2 shows that for $n = 200$, our models (Deep Norm (DN) and Wide Norm (WN) based Neural Metrics) achieve results comparable to that found by 400 iterations of TF in the symmetric case. We note that TF, which approaches the solution through the space of *all $n \times n$* matrices, produces a non-metric approximation, so that the number of **M3** violations ($\#_{\neg \mathbf{M3}}$) is greater than 0; this said, it is possible to fix these violations by adding a small constant to all entries of the matrix, which does not appreciably increase the loss $J_{MN}$. To test our models in the asymmetric case, we modified the TF algorithm; although our modified TF found a solution , our DN model performed significantly better. See Appendix D for complete experimental details.

Although triangle fixing is effective for small $n$, there is no obvious way to scale it to large datasets or to metric approximations of non-metric distance functions defined on a continuous domain. Using our models in these settings is natural, as we demonstrate in the next subsection.

## 3.3 APPLICATION: MODELING GRAPH DISTANCES

In the previous subsection we sought to "fix" a noisy, but small and fully observable metric; we now test the generalization ability of our models using large ($n > 100$K) metrics. We do this on the task of modeling shortest path lengths in a weighted graph $G = (\mathcal{V}, \mathcal{E})$. So long as edge weights are positive and the graph is connected, shortest path lengths are discrete quasi-metrics ($n = |\mathcal{V}|$), and provide an ideal domain for a comparison to the standard Euclidean approach.

Our experiments are structured as a supervised learning task, with inputs $x, y \in \mathcal{V}$ and targets $d(x, y)$. The targets for a random subset of 150K pairs (of the $O(|V|^2)$ total pairs) were computed beforehand

|          | $\vert\mathcal{V}\vert$ | $\vert\mathcal{E}\vert$ | $\max(d)$ | $\sigma_d$ | Sym? |          | Eucl. | WN | $\text{DN}_I$ | $\text{DN}_N$ | MLP |
|----------|------|------|------|------|------|----------|------|------|------|------|------|
| **to**   | 278K | 611K | 145.7 | 24.5 | $\leftrightarrow$ | **to**   | 12.5 | **6.6** | 6.7 | 6.7 | 12.3 |
| **3d**   | 125K | 375K | 86.7 | 13.2 | $\leftrightarrow$ | **3d**   | 31.2 | 17.3 | 15.4 | **12.9** | 20.6 |
| **taxi** | 391K | 752K | 111.2 | 13.4 | $\leftrightarrow$ | **taxi** | 14.4 | **10.6** | 11.8 | 11.4 | **5.8** |
| **push** | 390K | 1498K | 113.1 | 14.3 | $\rightarrow$ | **push** | 22.2 | 14.0 | 14.7 | **13.5** | 11.3 |
| **3dr**  | 123K | 368K | 86.5 | 13.1 | $\rightarrow$ | **3dr**  | 22.0 | **17.5** | 21.8 | 18.3 | 25.5 |
| **3dd**  | 125K | 375K | 97.8 | 13.4 | $\rightarrow$ | **3dd**  | 211.8 | 177.1 | 199.5 | **157.7** | 252.7 |

**(a)** Graph statistics      **(b)** Final test MSE @ $|D| = 50000$

**Table 4: Graph experiments. (a)** Statistics for different graphs. **(b)** Test MSE after 1000 epochs at training size $|D| = 50000$ (3 seeds). The best metric (and overall result if different) is bolded.

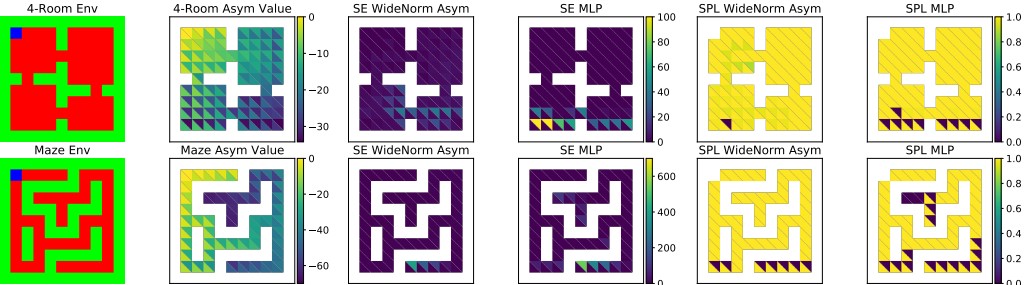

**Fig. 4: GVF environments. Left:** The **4-Room** (top) and **Maze** (bottom) environments, with walls (green), empty cells (red), and agent (blue). The next column shows example ground truth values, where state $s_0$ denotes agent at top left cell. Upper triangular regions indicate the value of $V(s_0, s)$ while lower indicate $V(s, s_0)$. **Center:** Squared Error (SE) heatmap between the learned and ground truth value for WideNorm and MLP architecture. **Right:** Success weighted by Path Length (SPL) metric heatmap. Higher is better. See Section 3.4 for detail and Appendix F for more visualizations.

using $A^*$ search and normalized to have mean 50. 10K were used as a test set, and the remainder was subsampled to form training sets of size $|D| \in \{1K, 2K, 5K, 10K, 20K, 50K\}$. Nodes were represented using noisy landmark embeddings (see Appendix E). We compare Wide Norms, Deep Norms (ICNN style, with ReLU activations, $DN_I$), and Deep Norm based Neural Metrics (with MaxReLU and concave activations, $DN_N$) to the standard Siamese-style Bromley et al. (1994) deep Euclidean metric and MLPs in six graphs, three symmetric and three not, summarized in Table 3a and described in Appendix E. Though MLPs do not induce proper metrics, they are expressive function approximators and serve as a relevant reference point. The results (shown for $|D| = 50K$ in Table 3b and expanded upon in Appendix E) show that our models tend to outperform the basic Siamese (Euclidean/Mahalanobis) network, and oftentimes the MLP reference point as well.

### 3.4 APPLICATION: LEARNING GENERAL VALUE FUNCTIONS

In this section, we evaluate our models on the task of learning a Universal Value Function Approximator (UVFA) in goal-oriented reinforcement learning (Sutton et al., 2011; Schaul et al., 2015). Figure 4 (left) illustrates the two 11x11 grid world environments, **4-Room** and **Maze**, in which we conduct our experiments. Each environment has a symmetric version, where the reward is constant (-1), and an asymmetric version with state-action dependent reward. We create a training set of transitions $(s, s', g, r, d)$ for the state, next state, goal, reward, and done flag, and the UVFA $V_\theta(s, g)$ is trained to minimize the temporal difference error via bootstrapped updates with no discounting. We tested several architectures at different training set sizes, which leaves out a portion of transitions such that only a fraction of states or goals were present during training. See Appendix F.1 for details.

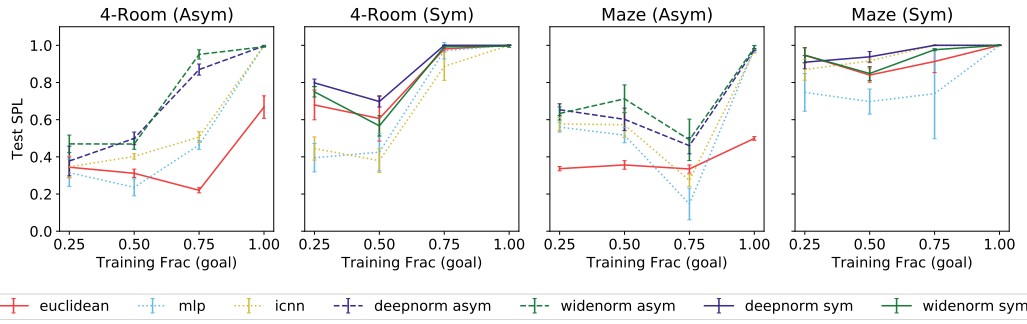

**Fig. 5: GVF results.** Generalization as measured by SPL metric (higher is better) on held out $(s, g)$ pairs as function of fraction of goals seen during training. Results averaged over 3 seeds and error bar indicates standard deviation. For fraction = 1 we evaluate on entire data.

Figure 5 summarizes the final generalization performance (on held out $(s, g)$ pairs) for each architecture after training for 1000 epochs. Performance is given in terms of the SPL metric (Anderson et al., 2018), which measures the success rate of reaching the goal weighted by the ratio of agent versus optimal path cost. We observe that when given only partial training data, Deep Norm and Wide Norm consistently outperform both MLPs and ICNNs at each sparsity level. As expected, the Euclidean metric could not solve the asymmetric environments. Qualitatively, we plot a heatmap of the squared difference (SE) between the learned and ground truth value function in Figure 4 (center), and observed that our proposed architectures have mostly lower SE than the MLP and ICNN architectures. Figure 4 (right) illustrates that WideNorm architecture is able to reach more cells (with almost optimal trajectory) in the environment when using a greedy policy with respect to the learned value function. Additional visualizations and results are in Appendix F.2.

## 3.5 COMPUTATIONAL CONSIDERATIONS

Several deep metric learning algorithms, such as semi-hard triplet mining Schroff et al. (2015), involve computing a pairwise distance matrix for large mini-batches. For example, Schroff et al. use mini-batches of size 1800. Computing this matrix for Euclidean distances can be done efficiently by taking advantage of the identity $\|x - y\|_2^2 = \|x\|_2^2 + \|y\|_2^2 - 2x^T y$ (see Section 4 of Oh Song et al. (2016) for details). This same identity can be applied to each component of a Wide Norm. There is no obvious way, however, to

|          | 32   | 128  | 512  | 2048 |
|----------|------|------|------|------|
| Euclidean | 0.18 | 0.27 | 0.45 | 1.06 |
| WN 3x600 | 1.59 | 1.57 | 1.75 | 2.36 |
| WN 64x64 | 15.7 | 13.4 | 17.7 | 26.3 |
| DN 2x400 | 0.97 | 5.73 | 76.9 | 293  |
| DN 3x600 | 1.50 | 11.4 | 174  | OOM  |

**Fig. 6:** Mean computation time (ms) for different mini-batch sizes (250 trials).

compute pairwise Deep Norms more efficiently than the naive $O(n^2)$ approach. To quantify, we recorded the pairwise distance computation time for our implementations (Table 6). Thus, although our previous experiments often found that Deep Norms performed slightly better (see, e.g., Figure 3 and Tables 2 and 3b), these results suggest that only Wide Norms are practical for large mini-batches.

## 4 DISCUSSION AND OTHER RELATED WORK

Deep Norms, Wide Norms, and Neural Metrics all respect the triangle inequality while universally approximating finite-dimensional asymmetric semi-norms. This allows them to represent metrics that the deep Euclidean Siamese architecture cannot, no matter how deep its embedding function is (see Figure 1). Our models thus provide a more expressive, non-Euclidean alternative for learning distances that satisfy the triangle inequality. As noted in the Introduction, this may useful for providing running time and error rate guarantees in clustering (Davidson and Ravi, 2009) and as an inductive bias to improve generalization performance (Figure 5; Hsieh et al. (2017)).

A line of theoretical work, surveyed by Indyk et al. (2017), characterizes the representational capacity of Euclidean space (and, more generally, $\ell_p$ space) by examining the asymptotic "distortion" of embedding $n$-point metric spaces into Euclidean space. Here the distortion of an embedding $\phi : \mathcal{X} \to \mathcal{X}'$ of metric space $(\mathcal{X}, d_{\mathcal{X}})$ into metric space $(\mathcal{X}', d_{\mathcal{X}'})$ is at most $c \geq 1$ if there exists $r > 0$ such that for all $x, y \in \mathcal{X}$, $r \cdot d_{\mathcal{X}}(x, y) \leq d_{\mathcal{X}'}(\phi(x), \phi(y)) \leq cr \cdot d_{\mathcal{X}}(x, y)$. The pioneering work of Bourgain (1985) bounds worst case distortion by $O(\log n)$, and Linial et al. (1995) shows that this bound is tight (i.e., $\Theta(\log n)$) for graph distances in $n$-point constant-degree expanders. This applies regardless of the dimensionality of the embedding space, and is true for all $\ell_p$ with $1 \leq p < \infty$. Future work might investigate the asymptotic distortion of our proposed architectures.

On account of the above limitations, others have also proposed non-Euclidean alternatives to learning and representing metrics. To improve expressivity, some have proposed non-parametric algorithms that learn distances directly. For instance, Biswas and Jacobs (2015) propose to frame clustering as a metric nearness problem and applying a quadratic programming algorithm; see also Gilbert and Jain (2017) and Veldt et al. (2018). Others learn parametric distances, as we do. Bi et al. (2015) parameterize and learn Cayley-Klein metrics, which are not translation-invariant. Yang and Jin (2006) and Nielsen et al. (2016) propose metrics that are similar to our Wide Norm, in that they use non-negative sums of several components (binary codes and Cayley-Klein metrics). As for symmetry, several papers have used asymmetric measures such as KL divergence (Vilnis and McCallum, 2014;

Vendrov et al., 2015; Chen et al., 2016; Ou et al., 2016). To our knowledge, we are the first to propose a parametric measure that is both asymmetric and satisfies the triangle inequality.

Another common approach in deep "metric" learning is to forgo the triangle inequality altogether and use non-metric similarity measures such as cosine similarity (Bromley et al., 1994; Yi et al., 2014). This can be sensible, as proper metrics are not always required. In image recognition, for example, the triangle inequality is questionable: why should $d(\text{CAT}, \text{WOLF}) \leq d(\text{CAT}, \text{DOG}) + d(\text{DOG}, \text{WOLF})$? In this case, Scheirer et al. (2014) find that non-metric similarities often perform better, concluding that "good recognition is non-metric". Thus, one should apply our models with care, in settings where the triangle inequality is thought to be a useful inductive bias (e.g., Hsieh et al. (2017)). Such applications we are excited about include learning search heuristics (Russell and Norvig, 2016) and scaling our UVFA models to more complex reinforcement learning problems (Plappert et al., 2018).

## 5 CONCLUSION

This paper proposed three novel architectures for modeling asymmetric semi-norms and metrics. They can be used instead of the usual Euclidean metric to increase expressiveness while respecting the triangle inequality. We showed that our models outperform Euclidean metrics when used with a Siamese-style deep metric learning architecture to (1) solve the metric nearness problem, (2) model shortest path lengths, and (3) learn general value functions in small reinforcement learning domains. Future work should explore larger scale applications such as facial recognition (Schroff et al., 2015), collaborative metric learning (Hsieh et al., 2017) and continuous control (Plappert et al., 2018).

### ACKNOWLEDGMENTS

We thank Cem Anil, James Lucas, Mitchell Stern, Michael Zhang and the anonymous reviewers for helpful discussions. Harris Chan was supported by an NSERC CGS-M award. Resources used in preparing this research were provided, in part, by the Province of Ontario, the Government of Canada through CIFAR, and companies sponsoring the Vector Institute (`www.vectorinstitute.ai/#partners`).

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

# A  PROOFS

In this Appendix, we restate each proposition from the main text and provide a short proof.

**Proposition 1.** *All positive homogeneous convex functions are subadditive; i.e., **C1** $\wedge$ **N2** $\Rightarrow$ **N3**.*

*Proof.* Putting $\alpha = 0.5$ in the definition of **C1** gives $f(0.5x + 0.5y) \leq 0.5f(x) + 0.5f(y)$. Applying **N2** on the left and multiplying by 2 gives $f(x + y) \leq f(x) + f(y)$ as desired. $\qquad\square$

**Proposition 2.** *If $\|\cdot\|$ is an asymmetric semi-norm, then $\|x\| = \|x| + \|-x|$ is a semi-norm.*

*Proof.* $\|x\| = \|x| + \|-x| = \|-x\|$, so **N4** is satisfied. **N2**-**N3** and non-negativity are similarly trivial. $\qquad\square$

**Proposition 3.** *If $\|\cdot\|_a$ is an (asymmetric) semi-norm, $\|\cdot\|_b$ is a norm (e.g., $\|\cdot\|_b = \|\cdot\|_2$), and $\lambda > 0$, then $\|x\|_{a+\lambda b} = \|x\|_a + \lambda\|x\|_b$ is an (asymmetric) norm.*

*Proof.* This follows from the positive definiteness of $\lambda\|x\|_b$ and the easily verified fact that non-negative weighted sums of semi-norms are semi-norms. $\qquad\square$

**Proposition 4.** *If $\|\cdot\|$ is an **N5** (semi-) norm on $\mathbb{R}^{2n}$, then $\|x| = \|\mathrm{relu}(x :: -x)\|$, where $::$ denotes concatenation, is an asymmetric (semi-) norm on $\mathbb{R}^n$.*

*Proof.* **N1** and **N2** are easily verified. To see that $\|\cdot|$ is not necessarily symmetric, choose $\|\cdot\|$ to be a Mahalanobis norm parameterized by a diagonal $W$ with $W_{ii} = i$ (but NB that if $\|\cdot\|$ is invariant to element-wise permutations, as are the $L_p$ norms, $\|\cdot|$ will be symmetric). For **N3**, we have:

$$
\begin{aligned}
\|x| + \|y| &= \|\mathrm{relu}(x :: -x)\| + \|\mathrm{relu}(y :: -y)\| \\
&\geq \|\mathrm{relu}(x :: -x) + \mathrm{relu}(y :: -y)\| \\
&\geq \|\mathrm{relu}(x + y :: -(x + y))\| \\
&= \|x + y|.
\end{aligned}
$$

The first inequality holds because $\|\cdot\|$ is **N3**. The second holds because $\|\cdot\|$ is **N5**, and we have either element-wise equality (when $\mathrm{sign}(x_i)$ and $\mathrm{sign}(y_i)$ agree) or domination (when they don't). $\qquad\square$

**Proposition 5.** *The Mahalanobis norm with $W = DU$, with $D$ diagonal and $U$ non-negative, is **N5**.*

*Proof.* Let $\|\cdot\|$ be $\|\cdot\|_2$, and $0 \leq x \leq y = x + \epsilon$. We have:

$$
\begin{aligned}
\|Wx\|\|Wy\| &\geq |(Wx)^T(Wy)| \\
&= xW^TW(x + \epsilon) \\
&= xW^TWx + xU^TD^TDU\epsilon \\
&\geq \|Wx\|^2.
\end{aligned}
$$

The first inequality is Cauchy-Schwarz. The second holds as all elements of $x$, $U^TD^TDU$, and $\epsilon$ are non-negative. $\qquad\square$

**Lemma 1** (Semilattice Stone-Weierstrass (from below)). *Let $C$ be a set of continuous functions defined on compact subset $K$ of $\mathbb{R}^n$, $L$ be a closed subset of $C$, and $f \in C$. If (1) for every $x \in K$, there exists $g_x \in L$ such that $g_x \leq f$ and $g_x(x) = f(x)$, and (2) $L$ is closed under $\max$ (i.e., $a, b \in L \Rightarrow \max(a, b) \in L$), then $\exists h \in L$ with $f = h$ on $K$.*

*Proof.* Fix $\epsilon > 0$ and consider the sets $U_x = \{y \mid y \in K, f(y) - \epsilon < g_x(y)\}$, for all $x \in K$. The $U_x$ form an open cover of $K$, so there is a finite subcover $\{U_{x_1}, U_{x_2}, \ldots, U_{x_n}\}$. Let $g = \max(g_{x_1}, g_{x_2}, \ldots, g_{x_n}) \in L$. We have $f - \epsilon \leq g \leq f$, and the result follows since $L$ is closed. $\qquad\square$

**Theorem 1** (UA for MICNNs). *The family $\mathcal{M}$ of Max Input Convex Neural Networks (MICNNs) that uses pairwise max-pooling (or MaxReLU) activations is dense in the family $\mathcal{C}$ of convex functions.*

*Proof.* For $f \in \mathcal{C}, x \in \mathbb{R}^n$, let $g_x \in \mathcal{M}$ be a linear function whose hyperplane in the graph of $f$ is tangent to $f$ at $x$. Then $g_x$ satisfies condition (1) of Lemma 1 (because $f$ is convex). The use of pairwise max activations allows one to construct $\max(h_1, h_2) \in \mathcal{M}$ for any two $h_1, h_2 \in \mathcal{M}$ by using $\log_2(n)$ max-pooling layers, satisfying condition (2) of Lemma 1. Thus $f$ is in the closure of $\mathcal{M}$, and the result follows. $\square$

**Theorem 2** (UA for Deep Norms and Wide Norms). *The families $\mathcal{D}$ of Deep Norms (using MaxReLU) and $\mathcal{W}$ of Wide Norms (using MaxMean) are dense in the family $\mathcal{N}$ of asymmetric semi-norms.*

*Proof.* The proof is almost identical to that of Theorem 1. The only subtlety is that $\mathcal{D}$ and $\mathcal{W}$ do not contain all linear functions in $\mathbb{R}^n$; they do, however, contain all linear functions whose hyperplane is tangent to any $f \in \mathcal{N}$, since $f$ is **N2**. This is easy to see for functions defined on $\mathbb{R}^2$. To generalize the intuition to $\mathbb{R}^n$, consider the ray $R_x = \{(\alpha x :: \alpha\|x\|) \mid \alpha \geq 0\} \subset \mathbb{R}^{n+1}$ defined for each $x \in \mathbb{R}^n$. By **N2**, this ray is a subset of the graph $g \subset \mathbb{R}^{n+1}$ of $f$. Furthermore, any hyperplane tangent to one point on this ray is tangent to the entire ray and contains all points on the ray, since the ray is linear from the origin—therefore the hyperplane contains the origin. But any hyperplane tangent to $g$ at $x$ is tangent to a point $(x)$ on the ray $R_x$, and so contains the origin. Since $\mathcal{D}$ and $\mathcal{W}$ contain *all* linear functions containing the origin, it follows that they contain all linear functions whose graph is tangent to $f$, in satisfaction of condition (1) of Lemma 1 (because $f$ is convex). For Deep Norms, the use of pairwise max activations allows one to construct a global max operation, as in the proof of Theorem 1, satisfying condition (2) of Lemma 1. Wide Norms using MaxMean have direct access to a global max, and so satisfy condition (2) of Lemma 1. It follows that $f$ is in the closures of $\mathcal{D}$ and $\mathcal{W}$. $\square$

**Proposition 6.** *The set of all asymmetric norms on $\mathbb{R}^n$ is in one-to-one correspondence with the set of all bounded and open convex sets ("unit balls") containing the origin.*

*Proof.* Given asymmetric norm $\|\cdot\|$ on $\mathbb{R}^n$, its unit ball $B_1 := \{x \mid \|x\| < 1\}$ is convex since, $\forall x, y \in B_1, \lambda \in [0, 1]$, we have $\|(1 - \lambda)x + \lambda y\| \leq (1 - \lambda)\|x\| + \lambda\|y\| \leq 1$ (using **N3** & **N2**). Conversely, given open and bounded convex set $B_1$ containing the origin, let $\|x\| = \inf\{\alpha > 0 \mid \frac{x}{\alpha} \in B_1\}$. **N1** and **N2** are straightforward and **N3** follows by noting that for $x, y \in \mathbb{R}^n$, $\alpha, \beta > 0$, such that $\frac{x}{\alpha}, \frac{y}{\beta} \in B_1$, we have $\frac{\alpha}{\alpha+\beta}\frac{x}{\alpha} + \frac{\beta}{\alpha+\beta}\frac{y}{\beta} = \frac{x+y}{\alpha+\beta} \in B_1$, so that $\inf\{\gamma > 0 \mid \frac{x+y}{\gamma} \in B_1\} \leq \inf\{\alpha > 0 \mid \frac{x}{\alpha} \in B_1\} + \inf\{\beta > 0 \mid \frac{y}{\beta} \in B_1\}$. $\square$

**Proposition 7** (Metric-preserving concave functions). *If $d : \mathcal{X} \times \mathcal{X} \to \mathbb{R}^+$ is (quasi-) metric, $f : \mathbb{R}^+ \to \mathbb{R}^+$, $f^{-1}(0) = \{0\}$, and $f$ is concave (i.e., $-f$ is convex), then $f \circ d$ is (quasi-) metric.*

*Proof.* This proposition is proven for metrics by Doboš (1998) (Chapter 1, Theorem 3). The extension to quasi-metrics is immediate, as the proof does not require symmetry. $\square$

**Proposition 8** (Max preserves metrics). *If $d_1$ and $d_2$ are (quasi-) metric, so too is $\max(d_1, d_2)$.*

*Proof.* **M1** and **M2** are trivial. For **M3**, let $d = \max(d_1, d_2)$. Given some $x, y, z$, we have both $d_1(x, y) + d_1(y, z) \geq d_1(x, z)$ and $d_2(x, y) + d_2(y, z) \geq d_2(x, z)$, so that $d(x, y) + d(y, z) \geq d_1(x, z)$ and $d(x, y) + d(y, z) \geq d_2(x, z)$. Therefore, $d(x, y) + d(y, z) \geq \max(d_1(x, z), d_2(x, z)) = d(x, z)$. Because if an element is larger than two other elements, it is also larger than their max. $\square$

**Erratum dated July 6, 2020** The originally published version of our paper did not cite the prior universal approximation result for ICNNs by Chen et al. (2018), which we were not aware of at the time. The text of Subsection 2.4 has been revised to reflect this prior work.

# B  IMPLEMENTATION DETAILS

Except where otherwise noted, these implementation details are common throughout our experiments.

Our Deep Norm implementation constrains each $W_i$ for $i < k$ to be non-negative by clipping the parameter matrix after each gradient update. For $W_k$, we use either a simple mean or MaxMean (see Subsection 2.3). We set $U_k = 0$. We use either ReLU or MaxReLU for our activations $g_i$, as noted.

Since the output layer is always scalar (size 1), we refer to Deep Norms in terms of their hidden layers only. Thus, a Deep Norm with $k = 3$ layers of sizes (400, 400, 1) is a "2x400" Deep Norm.

Our Wide Norm implementation avoids parameterizing the $\alpha_i$ in Equation 3 by absorbing them into the weight matrices, so that $\|x\| = \frac{1}{k} \sum_i \|U_i x\|_2$, where $U_i = \alpha_i k W_i$. Our asymmetric Wide Norm constrains matrix $U$ of Proposition 5 by clipping negative values and does not use matrix $D$ (i.e., we set $D = I$).

Neither our Deep Norm nor Wide Norm implementations impose positive definiteness. This simplifies our architectures, does not sacrifice representational power, and allows them to be used on pseudo-metric problems (where $d(x, y) = 0$ is possible for distinct $x, y \in \mathcal{X}$).

Neural Metrics involve only a very small modification to Deep Norms and Wide Norms: before applying the mean or MaxMean global activation to obtain the final output, we apply an element-wise concave activation function, as described in Subsection 3.1.

## C  2D NORMS ADDITIONAL DETAILS

**Dataset generation**  To generate data, we use Proposition 6 by generating a random point set, computing its convex hull, and using the hull as the unit ball of the generated norm, $\|\cdot\|$. Having obtained the unit ball for a random norm, we sample a set of $N = 500$ unit vectors in $\mathbb{R}^2$ (according to $L_2$) in random directions, then scale the vectors until they intersect the convex hull: $\{x\}^N$. We use these vectors for testing data, defined as $\mathcal{D}_{test} = \{(x^{(1)}, 1), ..., (x^{(N)}, 1)\}$. For training data, we sample a size $|D| = \{16, 128\}$ subset of these vectors and multiply them by random perturbations $\epsilon^{(i)} \sim \mathcal{U}(0.85, 1.15)$ to get $\{\hat{x} \,|\, \hat{x}^{(i)} = x^{(i)} \epsilon^{(i)}\}^k$. The training data is then defined as $\mathcal{D}_{train} = \{(\hat{x}^{(1)}, \epsilon^{(1)}), ..., (\hat{x}^{(k)}, \epsilon^{(k)})\}$.

**Models**  We compare 4 model types on 4 types of norms. The models are (1) Mahalanobis, (2) Deep Norms of depth $\in \{2, 3, 4, 5\}$ and layer size $\in \{10, 50, 250\}$, (3) Wide Norms of width $\in \{2, 10, 50\}$ and number of components $\in \{2, 10, 50\}$, and (4) unconstrained, fully connected neural networks with ReLU activations of depth $\in \{2, 3, 4, 5\}$ and layer size $\in \{10, 50, 250\}$ (MLPs). Although MLPs do not generally satisfy **N1-N4**, we are interested in how they generalize without the architectural inductive bias.

**Norms**  The norms we experiment on are random (1) symmetric and (2) asymmetric norms, constructed as above, and (3) square ($L_\infty$), and (4) diamond ($L_1$) norms.

**Training**  The models, parameterized by $\theta$, are trained to minimize the mean squared error (MSE) between the predicted (scalar) norm value $\|\hat{x}^{(i)}\|$ and the label norm value $\epsilon^{(i)}$: $\mathcal{L}(\theta) = \frac{1}{M} \sum_i^M (\|\hat{x}^{(i)}\|_\theta - \epsilon^{(i)})^2$, where $M = 16$ is the batch size. We use Adam Optimizer Kingma and Ba (2014), with learning rate 1e-3, and train for a maximum of 5000 epochs.

**Results**  Tables 5-7 show the test MSE for the best configurations for each target norm and training data size. Deep Norms performed orders of magnitude better than Mahalanobis and Wide Norms on the random symmetric and asymmetric norms, but Wide Norms performed better on the square and diamond norms. The MLP performed worse than our models, except on the asymmetric norm with small training data.

Figure 3 illustrates the learned norm balls for random symmetric and asymmetric norms, when trained with small ($k = 16$) and large ($k = 128$) data sizes. Appendix C includes additional visualizations. From the red contours, we observe that Deep Norms and Wide Norms generalize to larger and smaller norm balls (due to being **N2**), whereas the MLP is unable to generalize to the 0.5 norm ball.

### C.1  ADDITIONAL DETAILS ON NORM GENERATION

To generate point sets in $\mathbb{R}^2$, we generated $c \in [3, 10]$ clusters, each with $n_i \in [5, 50]$ random points. The points for each cluster were sampled from a truncated normal distribution with $\mu^i \sim \mathcal{U}(-0.5, 0.5)$ and $\sigma^{(i)} \sim \mathcal{U}(0.2, 0.6)$, truncated to 2 standard deviations. For symmetric sets, we considered only

| | Maha | DN | WN | MLP |
|---|---|---|---|---|
| **sym** | | | | |
| 16 | $3.4 \times 10^{-3}$ | $\mathbf{1.5 \times 10^{-5}}$ | $6.7 \times 10^{-5}$ | $1.5 \times 10^{-3}$ |
| 128 | $3.2 \times 10^{-3}$ | $\mathbf{7.8 \times 10^{-7}}$ | $1.0 \times 10^{-6}$ | $1.2 \times 10^{-5}$ |
| **asym** | | | | |
| 16 | $1.7 \times 10^{-1}$ | $\mathbf{4.1 \times 10^{-4}}$ | $4.9 \times 10^{-3}$ | $1.6 \times 10^{-3}$ |
| 128 | $1.7 \times 10^{-1}$ | $\mathbf{2.5 \times 10^{-5}}$ | $4.5 \times 10^{-3}$ | $3.0 \times 10^{-5}$ |
| **square** | | | | |
| 16 | $1.7 \times 10^{-2}$ | $3.0 \times 10^{-5}$ | $\mathbf{2.4 \times 10^{-9}}$ | $7.7 \times 10^{-3}$ |
| 128 | $1.1 \times 10^{-2}$ | $2.1 \times 10^{-6}$ | $\mathbf{9.3 \times 10^{-9}}$ | $7.8 \times 10^{-6}$ |
| **diamond** | | | | |
| 16 | $1.2 \times 10^{-2}$ | $6.5 \times 10^{-4}$ | $\mathbf{4.6 \times 10^{-8}}$ | $2.2 \times 10^{-3}$ |
| 128 | $1.1 \times 10^{-2}$ | $4.5 \times 10^{-7}$ | $\mathbf{2.3 \times 10^{-10}}$ | $7.8 \times 10^{-6}$ |

**Table 5:** MSE on 2d norm test set (target norm value = 1) for the best configuration of each type (allowing for early stopping), for each data size (16 and 128) and target norm.

| | Maha | DN | WN | MLP |
|---|---|---|---|---|
| **sym** | | | | |
| 16 | $1.4 \times 10^{-2}$ | $\mathbf{6.0 \times 10^{-5}}$ | $2.7 \times 10^{-4}$ | $3.8 \times 10^{-2}$ |
| 128 | $1.3 \times 10^{-2}$ | $\mathbf{3.1 \times 10^{-6}}$ | $4.0 \times 10^{-6}$ | $2.3 \times 10^{-2}$ |
| **asym** | | | | |
| 16 | $6.7 \times 10^{-1}$ | $\mathbf{1.6 \times 10^{-2}}$ | $2.0 \times 10^{-2}$ | $2.0 \times 10^{-1}$ |
| 128 | $6.7 \times 10^{-1}$ | $\mathbf{2.0 \times 10^{-4}}$ | $1.8 \times 10^{-2}$ | $2.8 \times 10^{-2}$ |
| **square** | | | | |
| 16 | $6.8 \times 10^{-2}$ | $1.2 \times 10^{-4}$ | $\mathbf{9.6 \times 10^{-9}}$ | $2.7 \times 10^{-1}$ |
| 128 | $4.4 \times 10^{-2}$ | $8.3 \times 10^{-6}$ | $\mathbf{3.7 \times 10^{-8}}$ | $6.1 \times 10^{-3}$ |
| **diamond** | | | | |
| 16 | $4.6 \times 10^{-2}$ | $2.6 \times 10^{-3}$ | $\mathbf{1.9 \times 10^{-7}}$ | $4.1 \times 10^{-1}$ |
| 128 | $4.4 \times 10^{-2}$ | $1.8 \times 10^{-6}$ | $\mathbf{9.3 \times 10^{-10}}$ | $7.4 \times 10^{-2}$ |

**Table 6:** MSE on 2d norm test set (target norm value = 2) for the best configuration of each type (allowing for early stopping), for each data size (16 and 128) and target norm.

| | Maha | DN | WN | MLP |
|---|---|---|---|---|
| **sym** | | | | |
| 16 | $8.5 \times 10^{-4}$ | $\mathbf{3.7 \times 10^{-6}}$ | $1.7 \times 10^{-5}$ | $4.5 \times 10^{-2}$ |
| 128 | $8.1 \times 10^{-4}$ | $\mathbf{2.0 \times 10^{-7}}$ | $2.5 \times 10^{-7}$ | $1.2 \times 10^{-2}$ |
| **asym** | | | | |
| 16 | $4.2 \times 10^{-2}$ | $\mathbf{1.0 \times 10^{-4}}$ | $1.2 \times 10^{-3}$ | $7.9 \times 10^{-2}$ |
| 128 | $4.2 \times 10^{-2}$ | $\mathbf{6.2 \times 10^{-6}}$ | $1.1 \times 10^{-3}$ | $7.6 \times 10^{-2}$ |
| **square** | | | | |
| 16 | $4.3 \times 10^{-3}$ | $7.5 \times 10^{-6}$ | $\mathbf{6.0 \times 10^{-10}}$ | $7.5 \times 10^{-2}$ |
| 128 | $2.7 \times 10^{-3}$ | $5.2 \times 10^{-7}$ | $\mathbf{2.3 \times 10^{-9}}$ | $1.0 \times 10^{-2}$ |
| **diamond** | | | | |
| 16 | $2.9 \times 10^{-3}$ | $1.6 \times 10^{-4}$ | $\mathbf{1.2 \times 10^{-8}}$ | $1.0 \times 10^{-1}$ |
| 128 | $2.8 \times 10^{-3}$ | $1.1 \times 10^{-7}$ | $\mathbf{5.8 \times 10^{-11}}$ | $2.5 \times 10^{-2}$ |

**Table 7:** MSE on 2d norm test set (target norm value = 0.5) for the best configuration of each type (allowing for early stopping), for each data size (16 and 128) and target norm.

those points with positive $x$-coordinates and also included their reflection about the origin. To ensure that the origin was inside the resulting convex hull, we normalized the points to have mean zero before computing the hull. The convex hull was generated by using the SciPy library Jones et al. (2001) `ConvexHull` implementation.

## C.2 Additional Visualizations

Figure 8 and Figure 7 further visualizes the training data and the Mahalanobis architecture in addition to the MLP, Deep Norm, and Wide Norm architectures, for the random symmetric, asymmetric, square ($L_\infty$) and the diamond ($L_1$) unit ball shape.

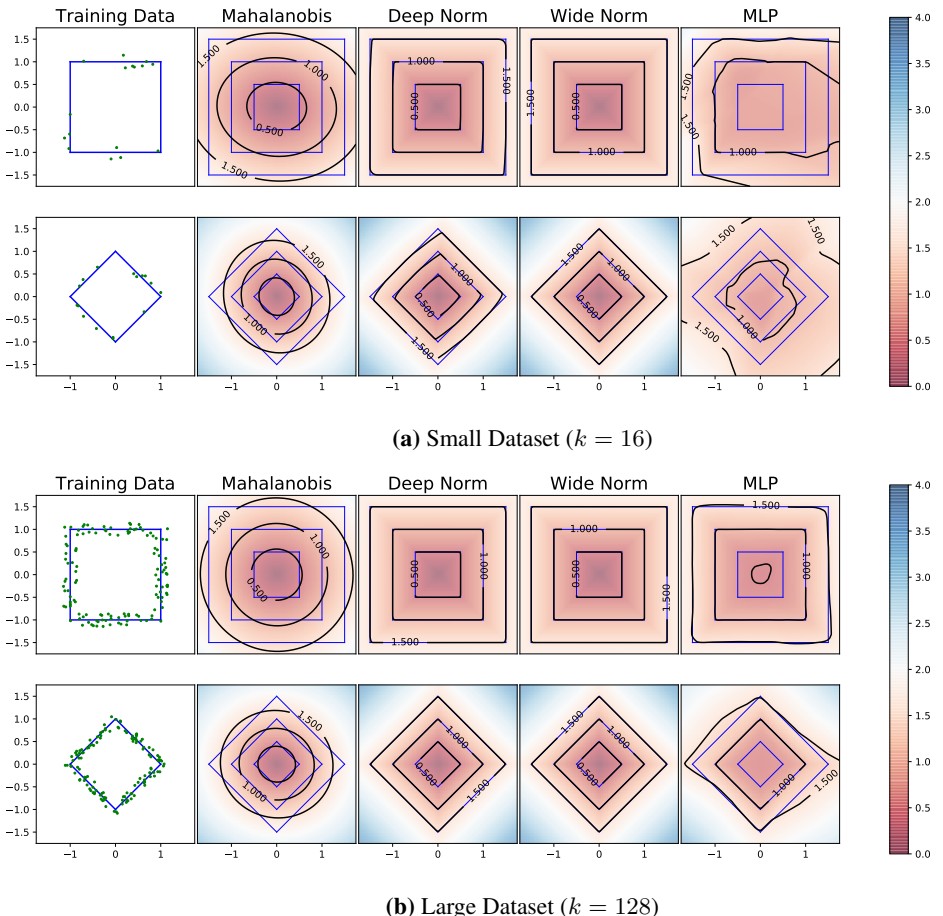

**Fig. 7:** Visualization of the 2d unit circles learned by several architectures (see Section 2.5) for a square and diamond shape convex hull, corresponding to $L_\infty$ and $L_1$ norm unit circle, respectively. The first column shows the training data points (in green), while the red line to the convex hull illustrates the portion of the convex hull which is covered by the training vector. Blue contours represent ground truth norm balls, and red contours learned norm balls, in each case for norm values $\{0.5, 1, 2\}$.

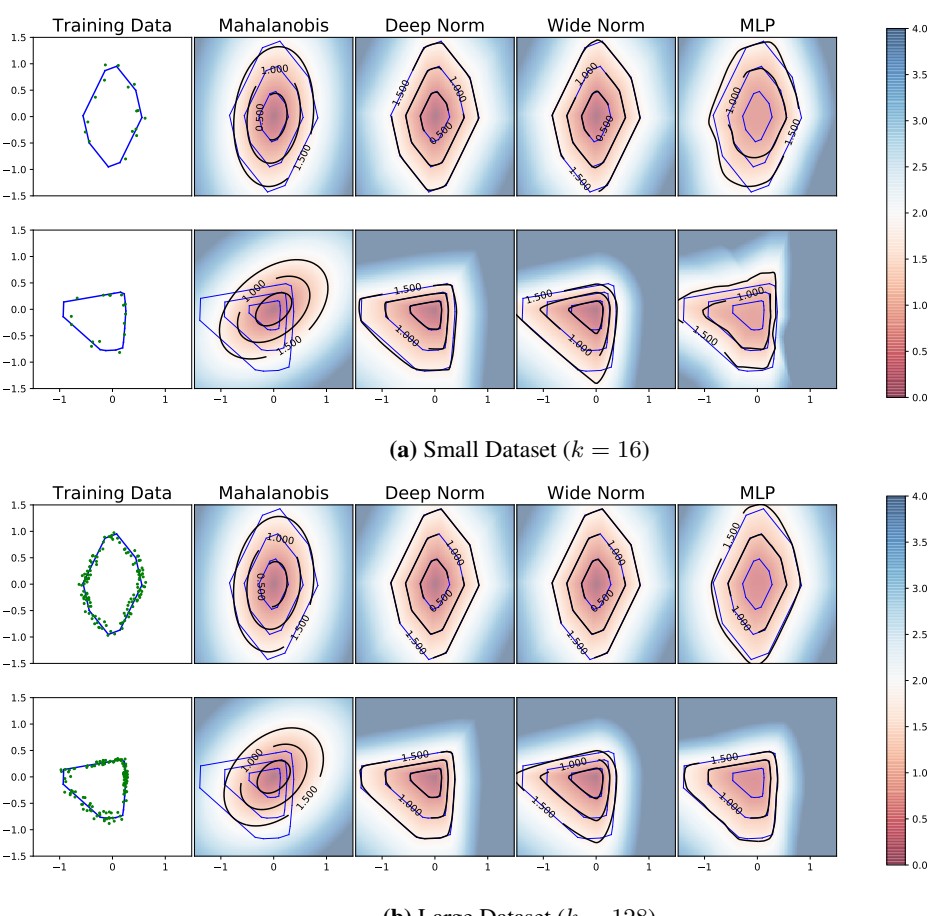

**(a)** Small Dataset ($k = 16$)

**(b)** Large Dataset ($k = 128$)

**Fig. 8:** Visualization of the 2d unit circles learned by several architectures (see Section 2.5) for a symmetric and asymmetric shape convex hull, corresponding to $L_\infty$ and $L_1$ norm unit circle, respectively. The first column shows the training data points (in green), while the red line to the convex hull illustrates the portion of the convex hull which is covered by the training vector. Blue contours represent ground truth norm balls, and red contours learned norm balls, in each case for norm values $\{0.5, 1, 2\}$.

## D   METRIC NEARNESS ADDITIONAL DETAILS

**Dataset creation**   For the symmetric metric nearness dataset, we generated the data as was found in Sra et al. (2005): a random matrix in $\mathbb{R}^{200 \times 200}$ was generated with values drawn from the uniform distribution ranging between 0 and 5. This matrix was added to its transpose to make it symmetric. Random uniform noise was added to each entry between 0 and 1, then the diagonal was removed. For the asymmetric case, a more complex dataset creation strategy was employed due to the fact that a random asymmetric matrix generated as above was too difficult a task for triangle fixing. Instead, we generated a random directed lattice graph with random weights generated from the exponential function applied to a random uniform sample between -1 and 1. The distances were calculated using Dijkstra's $A^*$ algorithm. Then, again, random uniform noise between 0 and 4 was added to the output, multiples of 10 removed to make the scale comparable to before, and the diagonals removed. The reason for the extra noise was due to the fact that the data matrix was already much closer to a metric.

**Architectures used**   For the symmetric metric nearness experiment, the Deep Norm based Neural Metric architecture used two layers of 512 neurons (we found that larger layer sizes learned much faster), with a 512-dimensional embedding function, MaxReLU activations, 5-unit concave activation functions, and a MaxMean global pooling layer. The Wide Norm based Neural Metric consisted of 128 Mahalanobis components of size 48, with a 512-dimensional embedding function, 5-unit concave activation functions, and MaxMean global pooling. The deep Euclidean architecture used a 1024-dimensional embedding function. For the asymmetric case, the architectures were the same except that we used the asymmetric equivalents for Deep Norm and Wide Norm.

**Training Regime**   The triangle fixing algorithm was allowed to go up to 400 iterations or convergence. For the symmetric case we reimplemented Sra et al. (2005) in Python using their C++ code as a template. For the asymmetric version, we used the same code but we removed their symmetrization step. For the training of all networks, we did 1500 epochs in total split up in to 500 epoch chunks were the learning rate decreased from 1e-3 to 3e-4 to 1e-4 at the end, using a batch size of 1000. We used the Adam optimizer Kingma and Ba (2014) with default hyperparameters. Results were averaged over 10 seeds.

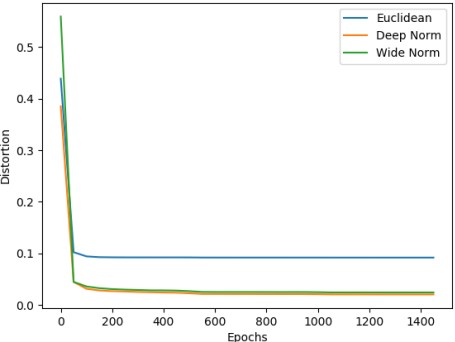

**Fig. 9:** Learning curves on symmetric metric nearness.

## E   GRAPH EXPERIMENTS ADDITIONAL DETAILS

**Symmetric graph descriptions**   The first symmetric graph, **to**, consists of a symmetrized road network extracted from openStreetMap (OSM, www.openstreetmap.org). The second, **3d**, represents navigation in a 50x50x50 cubic 3d gridworld, where the agent can move one step at a time in 6 directions, and movement wraps around the sides of the cube. The third, **taxi**, represents a 25x25 2d taxi environment, where the agent can move in four directions and there is a passenger that the agent can pick up and drop. Unlike in **3d**, there is no wraparound in **taxi**. Weights for edges in **to** correspond to the OSM distances, whereas weights for edges in **3d** and **taxi** are randomly sampled from $\{0.01, 0.02, \ldots, 1.00\}$ (but final distances are normalized so that the mean distance is 50).

**Asymmetric graph descriptions**   The first, **push**, is a 25x25 2d pusher environment, where the agent can move in four directions, and there is a box that the agent can push by moving into it. If the box is pushed into a wall, it switches places with the agent. The second, **3dd**, is a *directed* version of **3d**, where all paths in three of the six movement directions have been pruned (the same three directions for all nodes). The third, **3dr**, is a randomly pruned version of **3d**, where three random movement directions were pruned at each node, and any inaccessible portions of the resulting graph were pruned.

**Dataset generation**   Our experiments are structured as a supervised learning task, with inputs $x, y \in \mathcal{V}$ and targets $d(x, y)$. The targets for a random subset of 150K pairs (of the $O(|V|^2)$ total pairs) were computed beforehand using $A^*$ search and normalized to have mean 50. 10K were used as a test set, and the remainder was subsampled to form training sets of size $|D| \in \{1K, 2K, 5K, 10K, 20K, 50K\}$. Nodes were represented using noisy landmark embeddings. A subset of 32 landmark nodes was randomly chosen, and the distances to and from all other nodes in the graph were computed using Dijkstra's algorithm to form 32 base landmark features for symmetric graphs and 32 base landmark features for asymmetric graphs. These base landmark features were then normalized to have mean 0 and standard deviation 1, and noise sampled from $\mathcal{N}(0, 0.2)$ was added. To add additional noise through distractor features, 96 normally distributed features were concatenated with the noisy landmark features to obtain node embeddings with 128 dimensions for symmetric graphs and 160 dimensions for asymmetric graphs. We also tested node2vec embeddings Grover and Leskovec (2016), but found that our noisy landmark approach was both faster to run and produced results that were an order of magnitude better for all algorithms.

**Architectures**   We compare a 128-dimensional Mahalanobis metric (equivalent to a deep Euclidean metric with an additional layer), a Wide Norm based Neural Metric (32 components of 32 units, with 5-unit concave activations, and MaxMean global pooling), a plain Deep Norm (3 layers of 128 units, ICNN style with ReLU activations, no concave activations, and average pooling, $\text{DN}_I$), and a Deep Norm based Neural Metric (3 layers of 128 units, with MaxReLU and 5-unit concave activations, and MaxMean pooling, $\text{DN}_N$), and an MLP (3 layers of 128 units). We train each algorithm with 4 different embedding functions $\phi$, each a fully connected, feed-forward neural network with ReLU activations. The depths of tested $\phi$ ranged from 0 to 3 layers, all with 128 units. No regularization was used besides the size/depth of the layers.

**Training**   Training was done end-to-end with Adam Optimizer Kingma and Ba (2014), using an initial learning rate of 1e-3 and a batch size of 256. Networks were each trained for 1000 total epochs, and the learning rate was divided by 5 every 250 epochs.

**Results**   The complete results/learning curves are displayed in Figures 10 and 11 below.

## F   GENERAL VALUE FUNCTIONS ADDITIONAL DETAILS

### F.1   EXPERIMENTAL DETAILS

**4-Room and Maze environment description**   The fully observable grid environment has a state and goal representation of a binary tensor with dimensions $11 \times 11 \times 3$. Each cell in the 2D grid is represented by a 1-hot vector with 3 dimensions, indicating whether the cell is (1) empty, (2) wall, (3) agent. The agent can move up to 4 cardinal directions (North, South, East, West). If a wall is present in the direction then the agent cannot take that move. The 4-Room environment and Maze environment has a total of 4556 and 2450 training $(s, g)$ pairs, respectively. For each environment, the agent has access to a neighbour function $N(s)$ which returns a list of possible next states, corresponding to the empty cells adjacent to the current agent location. This is equivalent to having environment transition model $p(s'|s, a)$ over all the actions.

**Transition reward and episode termination**   For *symmetric* environments, every transition has a reward of -1, i.e., $R(s, s') = -1$. Note that reward is not a function of the goal, but the termination is. For *asymmetric* environments, we add a noise $\mu \sim \mathcal{U}(0, -5)$ to the base $-1$ reward for a transition if the direction of movement is west (left) or south (down). The done flag is set to 1 when the neighbouring state equals to the goal, i.e. $D(s, s', g) = [s' = g]$.

**Choosing Training-Test Split Dataset** Given the full training data set of $(s, g)$ pairs, we use 2 types of splitting into train and test datasets: (1) goal, and (2) state. We denote the *training fraction* $\eta =\in [0, 1]$ as the fraction of the total data points used for the training set. For goal splitting, we pick a subset of goals $\{g^*\} \subseteq g$ with $|\{g^*\}| \approx (1 - \eta)|\{g\}|$, and remove all $(s, g)$ pairs where $g \in g^*$, from the training set. These goals were chosen to be the percentage of empty cells from the bottom of the grid, when numbering the empty cells from left to right, and top-down (i.e. reading order of words in a page). This corresponds to approximately a fraction of the bottom rows of goals. In this case, the training set observes all possible states, and hence also all the transition rewards. We are interested in whether it can generalize to new unseen goals. For state splitting, we perform a similar procedure but instead with a subset of states which are removed, with the states corresponding to the agent being at the bottom rows of the grid. While all the goals are included in the training dataset, in the state splitting, some states and transition rewards are unknown to the agent, hence much more difficult to generalize.

**Architecture** The architectures used for the experiments consist of a shared feature extractor $\phi$ for the state $s$ and goal $g$, followed by a function $f_\theta$ on the difference of the features:

$$V(s, g; \phi, \theta) = f_\theta(\phi(g) - \phi(s)) \tag{4}$$

The feature extractor $\phi$ is composed of 2 convolutional layers with $3 \times 3$ kernel size, and 32 and 62 filters with ReLU activations and a stride of 1. We then flatten the feature maps and follow by 2 fully connected layers of hidden size 128 and 64, with ReLU activation on the first layer only. The variants of $f_\theta$ are summarized in table 8. Euclidean simply computes the L2 norm between the feature embeddings: $\|\phi(g) - \phi(s)\|_2$:

**Table 8:** Value Function Architectures

| $f_\theta$ | # Hiddens /Component Size | # Layers /Components | Act. Func. | Concave Units | Pool Func. |
|---|---|---|---|---|---|
| MLP | 64 | 3 | ReLU | - | - |
| ICNN | 64 | 3 | ReLU | - | - |
| DeepNorm | 64 | 3 | MaxReLU | 5 | Mean |
| WideNorm | 64 | 32 | - | 5 | Mean |

**Training** The objective function for training the models is the Temporal Difference (TD) Error $L(\phi, \theta)$:

$$L(\phi, \theta) = \mathbb{E}_{(s,g) \sim \mathcal{D}}\left[\left(V(s, g; \phi, \theta) - y\right)^2\right], \tag{5}$$

$$y = \begin{cases} r(s, s', g) + \max_{s' \in N(s)} V(s', g; \bar{\phi}, \bar{\theta}), & \text{if } s' \neq g \\ r(s, s', g), & \text{otherwise} \end{cases} \tag{6}$$

Where $N(s)$ refers to the set of next states of $s$ after applying different actions at that state. Note that we did not apply a discount factor on the value of the next state (i.e. the discount factor $\gamma = 1$). We make use of *target* networks $\bar{\phi}, \bar{\theta}$ when computing the target $y$. The target networks are updated once every epoch of training, via exponential moving average (Polyak Averaging) with the main networks $(\phi, \theta)$:

$$\bar{\theta}^{(n+1)} = \alpha \bar{\theta}^{(n)} + (1 - \alpha)\theta^{(n)}, \tag{7}$$

where, $\alpha = 0.95$ is the update fraction. We use Adam Kingma and Ba (2014) with learning rate 0.0001, batch size 128, for 1000 epoch. We evaluate SPL metric every 200 epoch.

**Evaluation Metrics** We evaluate the learned value functions on several metrics: MSE, Policy Success, and SPL Anderson et al. (2018). The MSE was calculated on on the ground truth value of the held out test set $V(s, g)$. For policy success and SPL, we select $N = 100$ random $(s, g)$ pairs in the test set, initialize the agent at $s$, and follow a greedy policy to visit the neighbour $s' = \arg\max'_s r(s, s') + V(s')$ for up to $T = 121$ timesteps ($11 \times 11$). If the agent reaches the desired goal within the episode $i$ then the binary success indicator $S_i = 1$ and the episode terminates, and 0 otherwise. Let $l_i$ be the ground truth $V(s, g)$ optimal episode cumulative reward (i.e. path cost), and $p_i$ the cumulative reward in the episode by the agent's trajectory. We then define the Policy Success rate and SPL as:

$$Success = \frac{1}{N} \sum_{i=1}^{N} S_i, \quad SPL = \frac{1}{N} \sum_{i=1}^{N} S_i \frac{l_i}{\max(p_i, l_i)} \tag{8}$$

## F.2 ADDITIONAL EVALUATIONS AND VISUALIZATIONS

We visualize additional heatmaps for (1) learned value function (Figure 15), (2) Squared Error (SE) between the ground truth value and the learned value (Figure 16), and (3) SPL metric for individual $(s_0, s)$ and $(s, s_0)$ pairs (Figure 17). These results are for agents trained with training fraction $\eta = 0.75$.

In comparing the performance versus training fraction, we also plot the final performance after 1000 epoch of training: (1) training set SPL versus training fraction (Figure 13a), (2) Train (Figure 14a) and test (Figure 14b) mean squared error (MSE) with the groundtruth values. We note that while our Deep Norm and Wide Norm appears to have higher MSE than the baselines, in practice when utilizing the value function with greedy policy, our architectures were able to achieve better success than the baseline architectures.

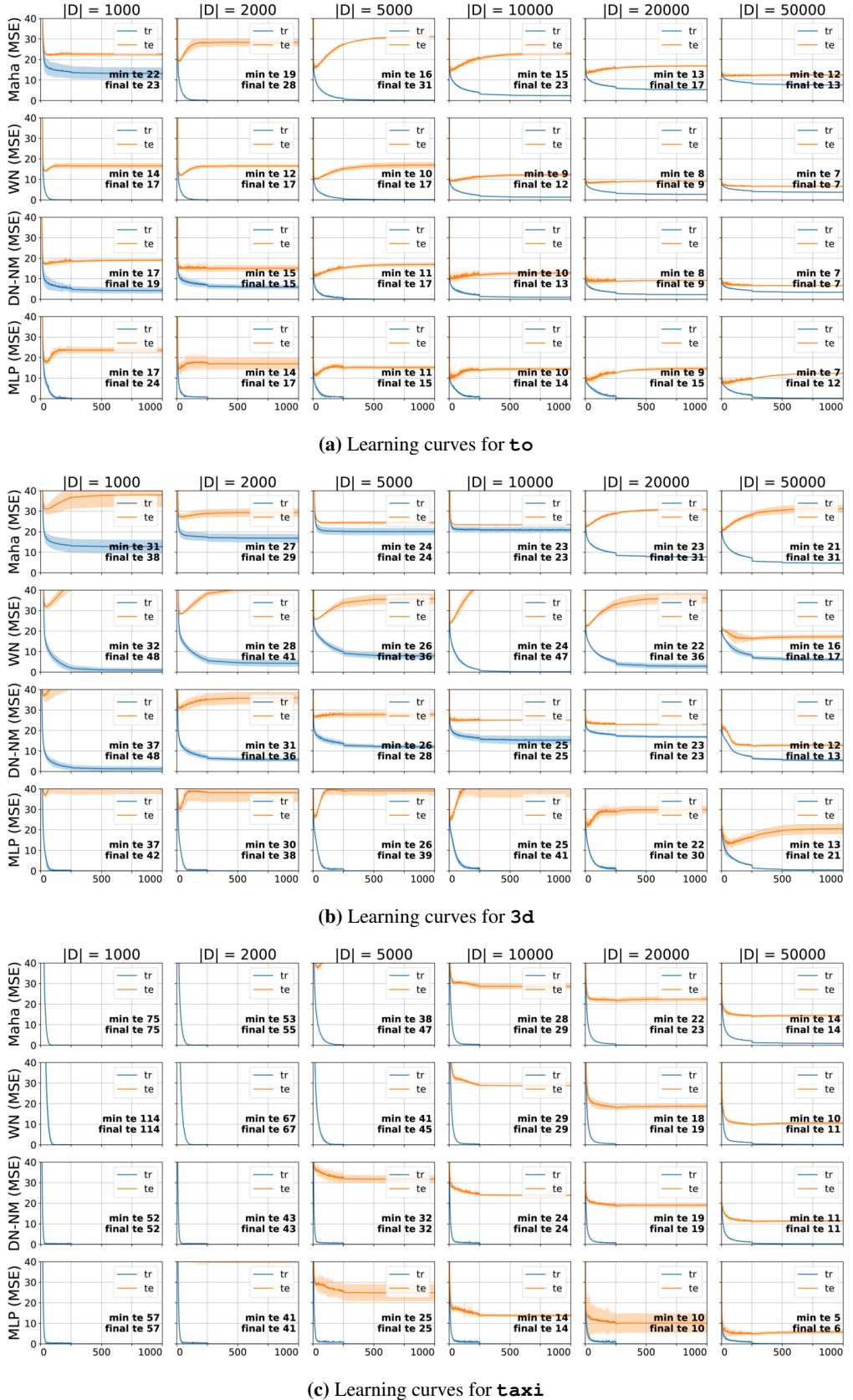

(a) Learning curves for `to`

(b) Learning curves for `3d`

(c) Learning curves for `taxi`

Fig. 10: Symmetric graph learning curves.

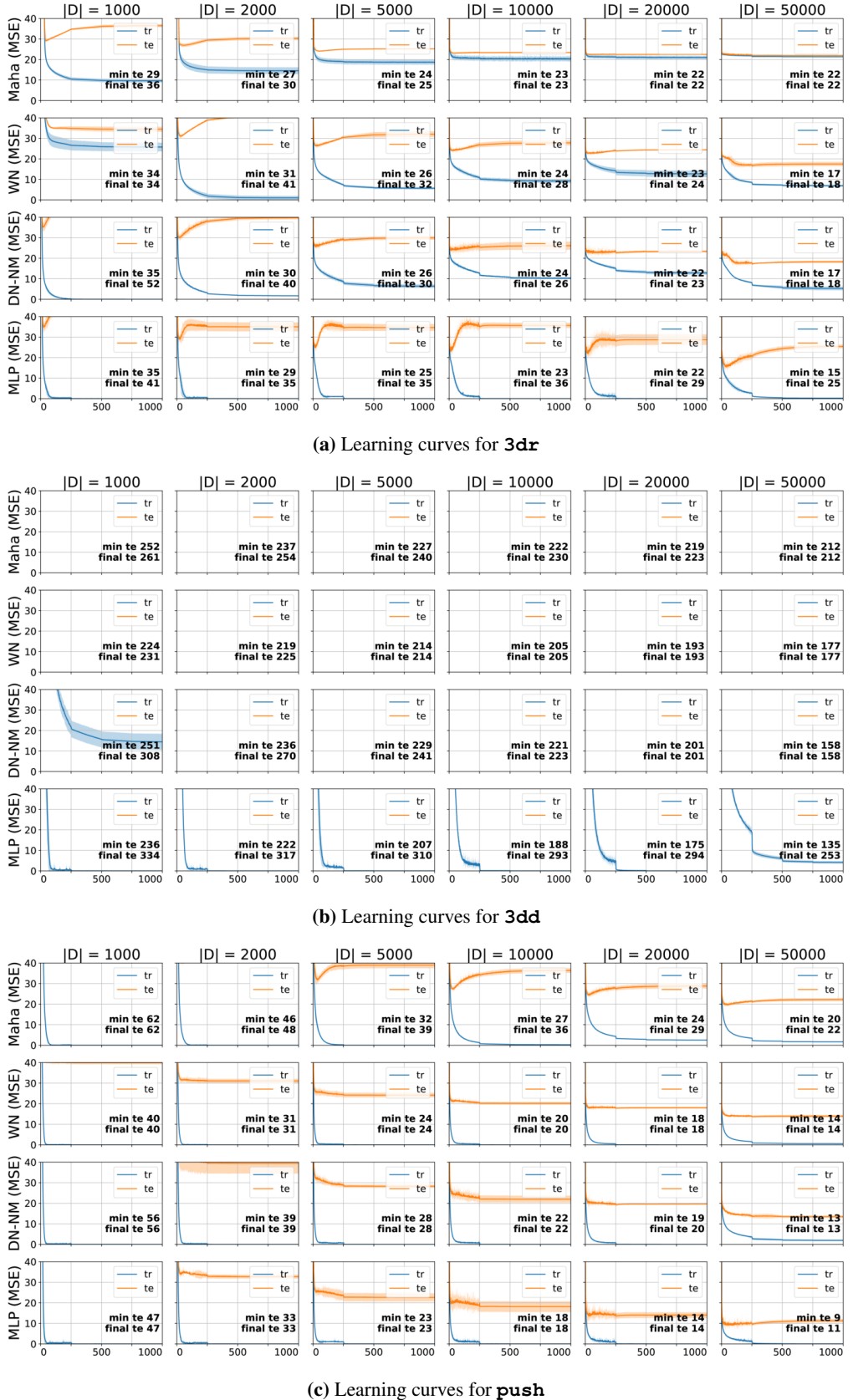

(a) Learning curves for **3dr**

(b) Learning curves for **3dd**

(c) Learning curves for **push**

**Fig. 11:** Asymmetric graph learning curves.

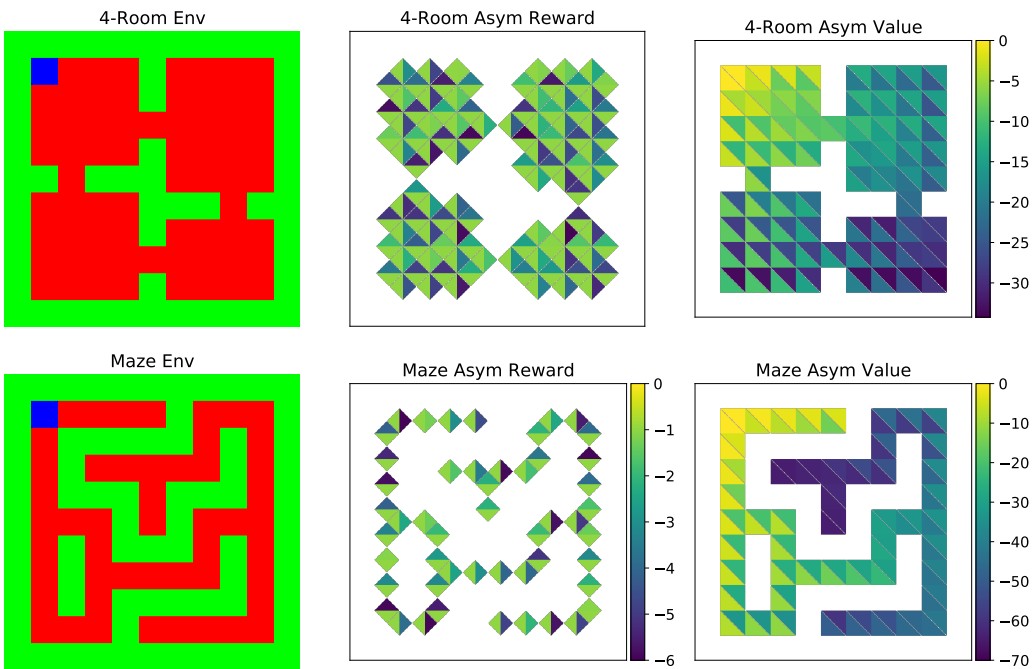

**Fig. 12: GVF environments. Left:** The `4-Room` and `Maze` environments, with walls (green), empty cells (red), and agent (blue). **Center:** Rewards in asymmetric environments. Symmetric versions have reward -1 everywhere. **Right:** Example ground truth value, where state $s_0$ denotes agent at top left cell. Upper triangular regions indicate the value of $V(s_0, s)$ while lower indicate $V(s, s_0)$.

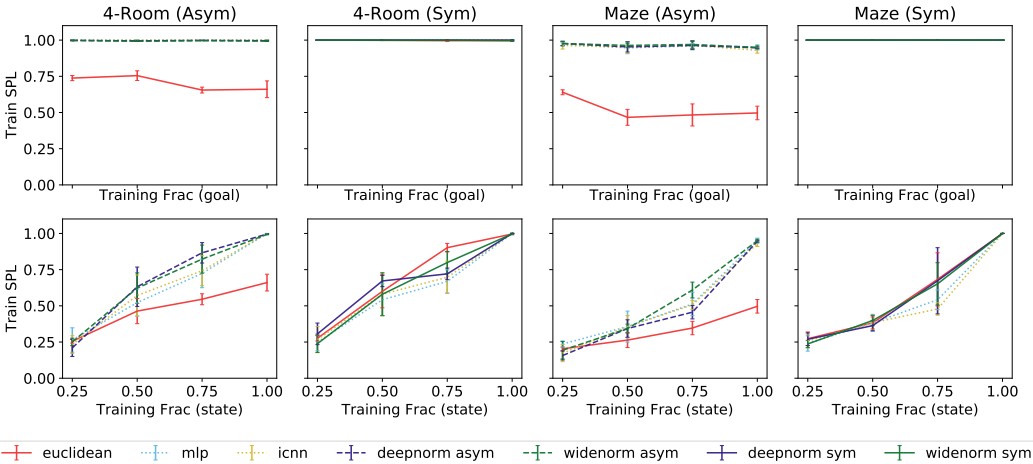

(a) Train Set SPL versus fraction of training set.

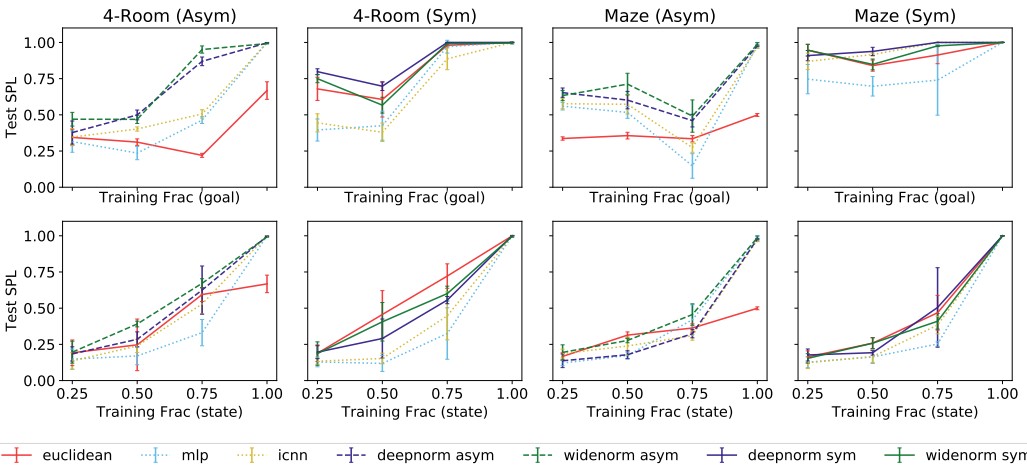

(b) Test Set SPL versus fraction of training set.

**Fig. 13:** Additional performance on the train/test SPL metric vs training fraction for the four environment variants (higher is better).

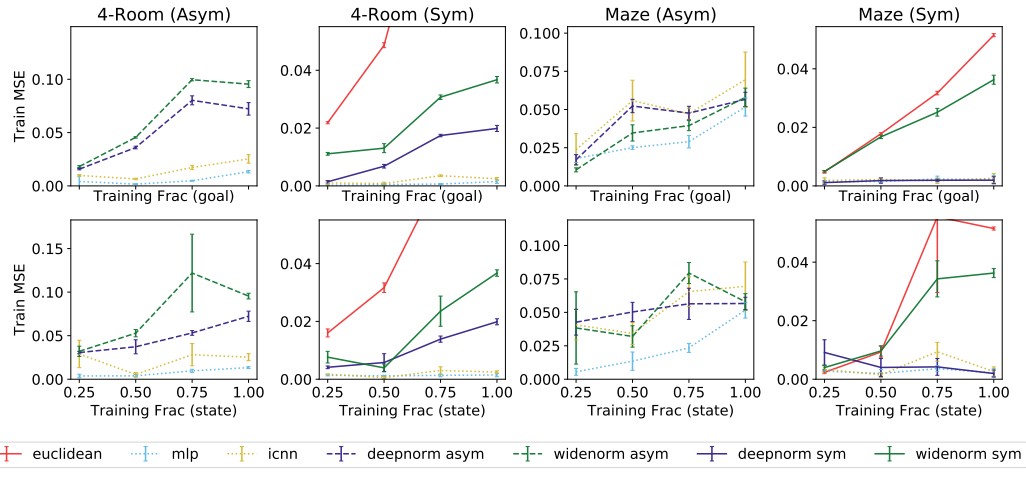

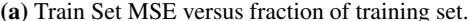

**(a)** Train Set MSE versus fraction of training set.

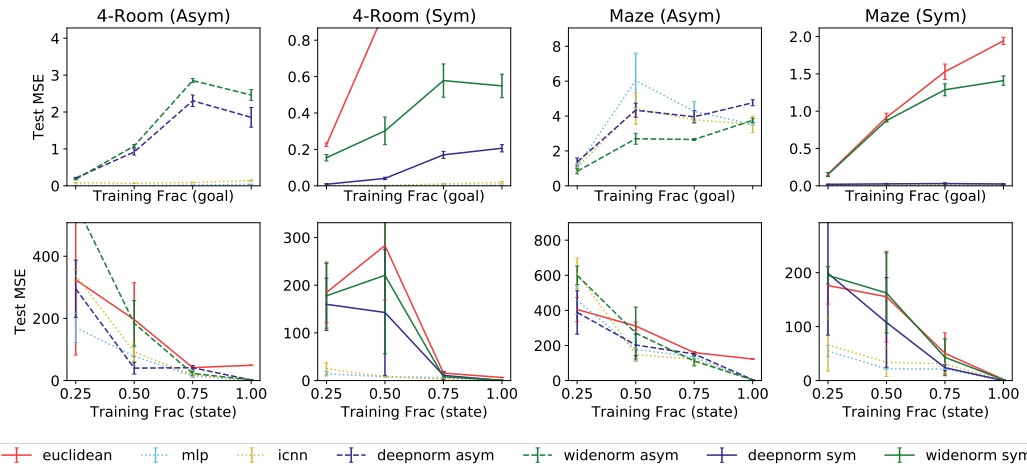

**(b)** Test Set MSE versus fraction of training set.

**Fig. 14:** Additional performance on the train/test MSE metric vs training fraction for the four environment variants (lower is better).

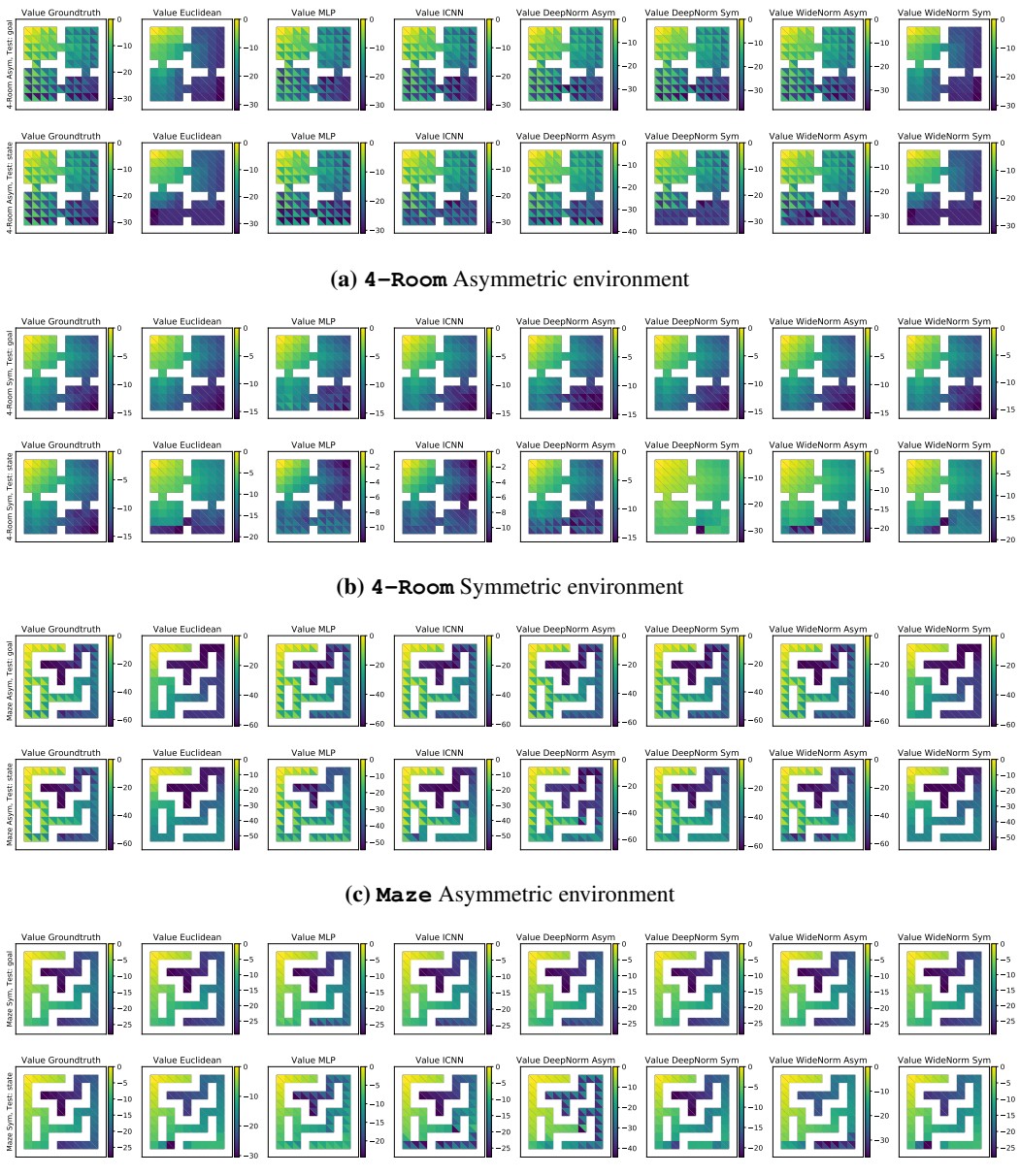

(a) **4-Room** Asymmetric environment

(b) **4-Room** Symmetric environment

(c) **Maze** Asymmetric environment

(d) **Maze** Symmetric environment

**Fig. 15:** Learned Value Function for the four environments with Train Fraction = 0.75.

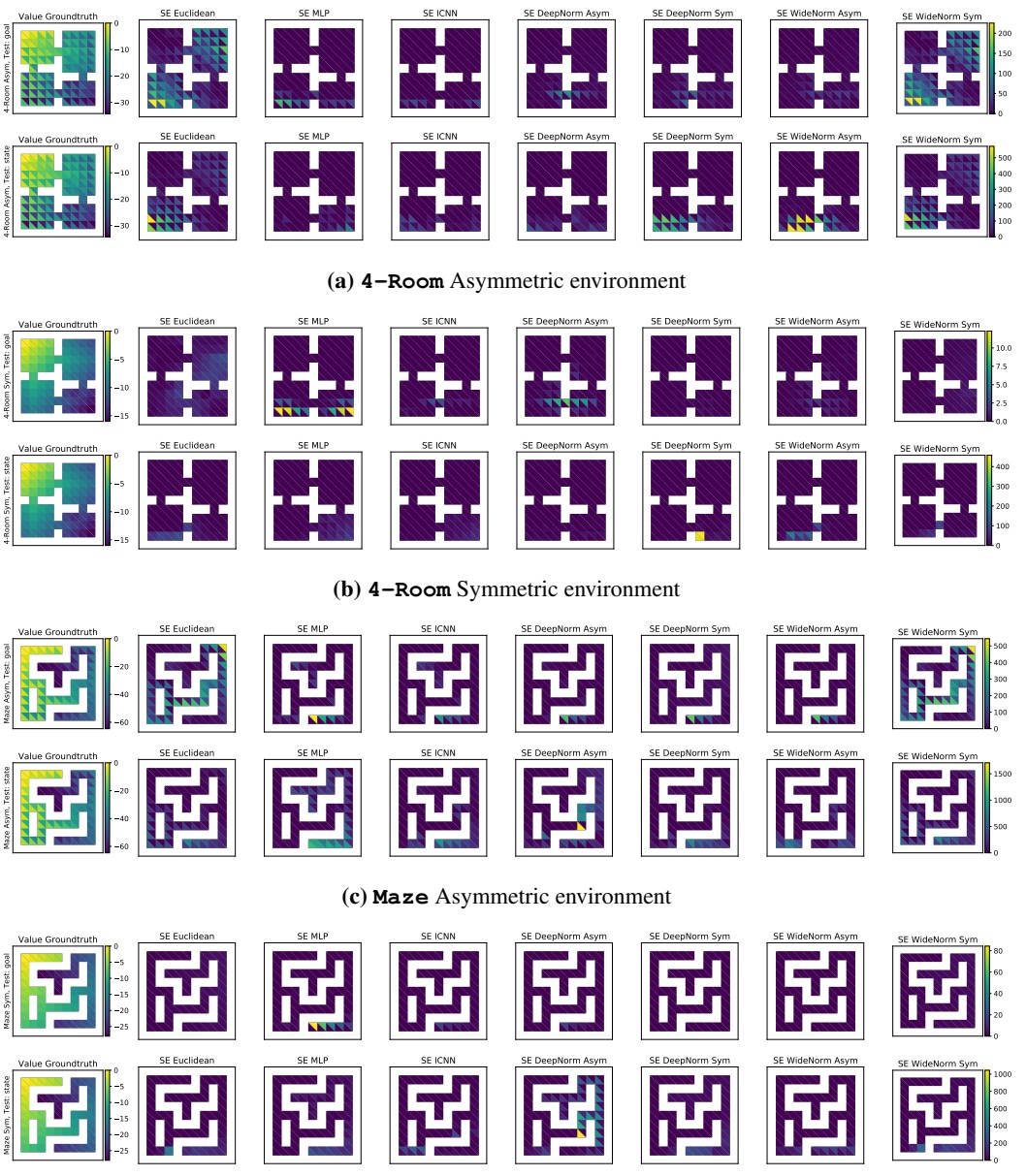

**Fig. 16:** Squared Error (SE) compared to the ground truth value function for the four environments with Train Fraction = 0.75.

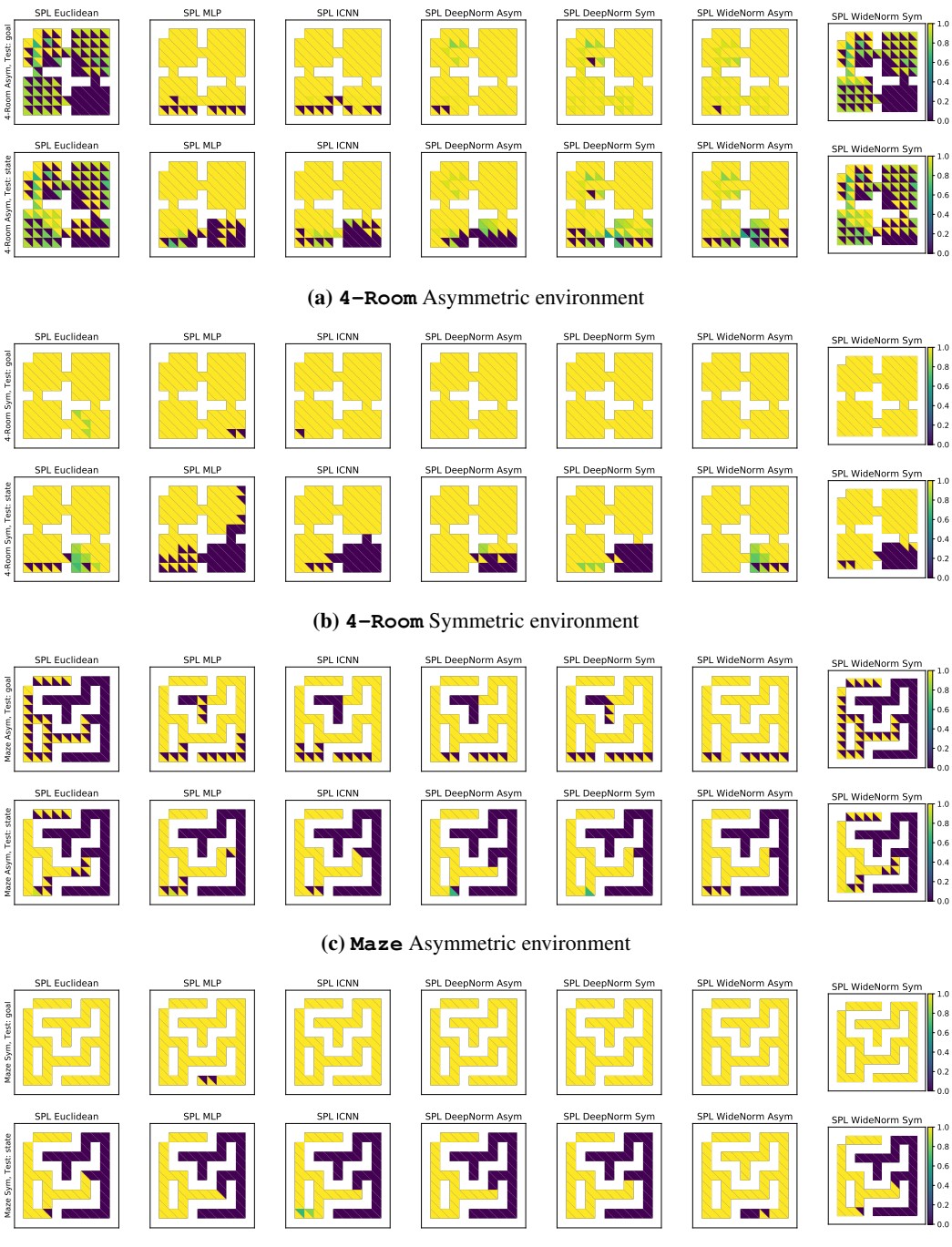

**Fig. 17:** Success weighted by Path Length (SPL) metric the four environments with Train Fraction = 0.75.

