# OpenReview forum: "An Inductive Bias for Distances: Neural Nets that Respect the Triangle Inequality"
_ICLR.cc/2020/Conference — Accept (Poster)_

### Official Review · AnonReviewer1 · 2019-10-25
**Official Blind Review #1**

**Rating:** 8

**Review:**

This paper shows how to enforce and learn non-Euclidean
(semi-)norms with neural networks.
This is a promising direction for the community as part
of a larger direction of understanding how to do better
modeling in domains that naturally have non-Euclidean
geometries.
This paper shows convincing experiments for modeling
graph distances and in the multi-goal reinforcement
learning setting.

One clarification I would like is related to some
tasks inherantly not having a Euclidean embedding.
Are there works that theoretically/empirically characterize
how bad this assumption can be for some problem classes?
Even though some tasks are impossible to embed exactly
into a Euclidean space, are there sometimes properties
that, given a high enough latent dimensionality,
they can be reasonably approximated?
And in the table of Figure 1, whta dimension n was used
for the Euclidean norm?

**Experience Assessment:**

I have read many papers in this area.

**Review Assessment: Checking Correctness Of Derivations And Theory:**

I assessed the sensibility of the derivations and theory.

**Review Assessment: Checking Correctness Of Experiments:**

I assessed the sensibility of the experiments.

**Review Assessment: Thoroughness In Paper Reading:**

I read the paper at least twice and used my best judgement in assessing the paper.

---

> ### Author Response · Authors · 2019-11-09
> **Author Response to Reviewer 1**
>
> Thank you for taking the time to review our paper and for your questions regarding the expressiveness of Euclidean embeddings. As noted by Reviewer 3, there is a rich literature on this topic. We will aim to reference some of the relevant works in an update to our paper. Perhaps the main results to be aware of, which we think are responsive to your query, are displayed in Rows 1, 2 and 4 of Table 8.4 of Indyk et al.’s chapter on metric embeddings [1]. The result referenced in Row 1 says that there is an $O(\log n)$ upper bound on distortion (defined on pdf pg 2 of [1]) when embedding an $n$-point metric space into Euclidean space. The Row 2 result says that for some types of metric spaces (constant degree expanders, as noted by Reviewer 3), the $O(\log n)$ bound is tight, even if we let the latent dimensionality go to infinity. The Row 4 result says that any metric space can be embedded into $\ell^d_\infty$ given large enough latent dimensionality (the construction, known as Frechet’s embedding, is given at the bottom of pdf page 4 of [1]). Since our architectures approximate $\ell^d_\infty$ (see, e.g., Fig 7 on pg 17 of our submission), we inherit this result. We’re happy to discuss this further.
>
> For the table in Figure 1, note that 4 embedded points occupy a subspace of dimensionality no greater than 3, so that additional latent dimensionality would not add to potential expressiveness; nevertheless we did test this with various dimensions >= 4 (e.g., d=32) in case it was easier to optimize, and consistently obtained the same result.
>
> [1] Indyk, Matousek, Sidiropoulos, Low-Distortion Embeddings of Finite Metric Spaces (2017). Available here: http://www.csun.edu/~ctoth/Handbook/chap8.pdf

---

> > ### Comment · AnonReviewer1 · 2019-11-15
> > **Response**
> >
> > Thanks for the response and additional reference! I've read over the other reviews and responses and still maintain my original score of an accept.

---

### Official Review · AnonReviewer4 · 2019-10-28
**Official Blind Review #4**

**Rating:** 3

**Review:**

This paper is about learning and utilizing distance metrics in neural nets, particularly focusing on metrics that obey the triangle inequality. The paper's core contributions are three approaches to doing this (Deep Norms, Wide Norms, Neural Metrics), along with theoretical and empirical grounding for the metrics.

The hypothesis is clearly on the bottom of page 1 - "Is it possible to impose the triangle inequality architecturally, without the downsides of Euclidean distance?" The downsides are previously listed as 1) not being able to represent asymmetric distances and 2) that the Euclidean space is known to not be able to precisely model some metric spaces with an embedding.

The approach given is quite well motivated. At large, this paper is quite clear and does a good job of delineating why it is taking each step. That starts with a preliminary discussion of metric spaces and norms and what we get when we have the given properties in different combinations.

After describing the motivations and the differences between the three algorithms, the paper then goes on to show results on a couple of toy tasks (metric nearness, graph distances) and then a more challenging one in learning a UVFA. The most striking of the results is the UVFA one where all of the metrics do much better than the Euclidian norm on the asymmetric case, which is the usual one. If these results held in over bigger environments and/or much more data, that would be really intriguing.

I do feel as if this paper is missing a glaring experiment. It talks a lot at the beginning about Siamese Networks being a motivation. It then doesn't do anything with Siamese Networks. They are very common and, if the Euclidean Metric was really deficient relative to this one, we would see it in those results given how important is the relative differences of the embeddings in Siamese Networks.

We also don't see an example where the Euclidean metric fails to come even close to the other metrics in efficacy (as appealed to in the second downside for the Euclidean metric). I don't think that the UVFA results are this because they are cut quite short - it could just be an artifact of the learning process being slow a la how SGD is frequently shown to be just as good as more complex optimizers given the right tuning.

Finally, while I do work on some areas of representation learning, this is not my forte and so I'm not too familiar with most results in this domain. That being said, I am not entirely convinced that this result is of huge consequence unless the empirical analysis is strengthened a lot. Examples of that would include the two I described above.

In its current form, I do not think that this passes the ICLR bar, however I do think its close and, if the experiments I prescribed continued the trend, I would suggest its inclusion.

**Experience Assessment:**

I have read many papers in this area.

**Review Assessment: Checking Correctness Of Derivations And Theory:**

I did not assess the derivations or theory.

**Review Assessment: Checking Correctness Of Experiments:**

I assessed the sensibility of the experiments.

**Review Assessment: Thoroughness In Paper Reading:**

I read the paper at least twice and used my best judgement in assessing the paper.

---

> ### Author Response · Authors · 2019-11-09
> **Author Response to Reviewer 4**
>
> Thank you for taking the time to review our paper and your suggestions on improving the empirical portions of our paper.
>
> In regards to the comment that our paper does not do anything with Siamese Networks, we would like to clarify that all baselines marked “Euclidean” (or “Mahalanobis”) are Siamese networks: the same embedding function (shared parameters, i.e. Siamese-style) is used to embed both source / target points into the latent Euclidean space. In this sense, our own architectures are also Siamese networks. We will try to clarify this in an update.
>
> In regards to your comments on showing the relative deficiency of the Euclidean metric, we think the symmetric graph experiments show that our architectures are consistently more expressive than the Euclidean Siamese networks: if you compare the top row of each Subfigure on Page 22 of the Appendix (Euclidean Siamese Network) to the 3rd row (Deep Norm), the DN training performances (blue curves) are consistently better, and generalization performance (orange curves) are better except for the low data case on 3d, which can be attributed to the Euclidean norm acting as a regularizer to prevent overfitting. We note that these are fairly sizable graphs, of over 100,000 nodes each, so that this task might be considered more challenging than the UVFA task; indeed, none of the tested architectures obtained particularly good performance on the “3dd” graph.
>
> In regards to your comment on our UVFA experiment, could you please clarify what you mean by “cut quite short”? For these experiments we ran 1000 epochs over the training set, which was sufficient for the loss to visibly converge: you may note from Figure 5 that when all data is provided, all architectures (except Euclidean in the asymmetric case) achieve perfect SPL, which indicates that we have trained to convergence.

---

### Official Review · AnonReviewer3 · 2019-10-30
**Official Blind Review #3**

**Rating:** 8

**Review:**

This paper proposes a modeling approach for norm and metric learning that ensures triangle inequalities are satisfied by the very design of the architecture. The main idea is that convexity together with homogeneity imply subadditivity, so starting from an input-convex architecture and using activations that preserve homogeneity implies the resulting model is sub-additive at every point. This architecture is used to model a norm, and in conjunction with an embedding - a metric. The authors also propose a mixture-based approach that combines a given set of metrics into a new one using a max-mean approach. Universal approximation results are presented for both architectures. The results are illustrated on a few mostly synthetic examples including metric nearness for random matrices, value functions for maze MDPs and distances between nodes on a graph (some problems here are sourced from open street map).

I think this is one of those papers where there is nothing much to complain about. I found the paper to be very-well written. The basic modeling approach of propagating homogeneity through an input-convex net is elegant, and conceptually appealing.

My only suggestion to the authors is that it looks as if a lot of importance is placed on modeling asymmetry, however, that problem seems relatively easily solvable with existing approaches. One could just have two separate embedding functions for the two positional arguments. I don't know if there are obvious reasons why this wouldn't work, but it looks like a very sensible idea that could solve asymmetry. I think that the other issue, that of nonembeddability is much more important, yet it was not emphasized particularly strongly except for one example. I think expanding on this would strengthen the motivation significantly. There is a rich literature on (non)embeddability in l2, which contains some deeply non-intuitive results (e.g. large graph classes like expanders being non-embeddable). I think that a quick survey on that citing the most important results would make the seriousness of this issue apparent to the reader.

**Experience Assessment:**

I have read many papers in this area.

**Review Assessment: Checking Correctness Of Derivations And Theory:**

I assessed the sensibility of the derivations and theory.

**Review Assessment: Checking Correctness Of Experiments:**

I assessed the sensibility of the experiments.

**Review Assessment: Thoroughness In Paper Reading:**

I read the paper thoroughly.

---

> ### Author Response · Authors · 2019-11-09
> **Author Response to Reviewer 3**
>
> Thank you for taking the time to review our paper and for the suggestion to include a discussion on non-embeddability. We agree it would be good to do so, and will aim to provide this in an update to the paper (see also our response to Reviewer 1, who had some questions on this topic).
>
> Regarding asymmetry, in our literature review we did not find any existing approach to enforcing the triangle inequality in an asymmetric setting. The idea you propose of separate embedding functions came up in early discussions (it was also proposed in the supplementary material of Schaul et al. to deal with asymmetry in RL — see $D_S$ on pg 2 of [1]). While this is certainly asymmetric (as would be any two-argument MLP), it does not satisfy our goal of enforcing triangle inequality architecturally. Since our source node embedding $\phi(x)$ is not necessarily equal to the goal node embedding $\psi(x)$, we can pick $\phi, \psi$ such that $d(y, x) + d(x, z) < d(y, z)$ by having $\phi(y)$ embed close to $\psi(x)$ and $\psi(z)$ embed close to $\phi(x)$. Note also that using Euclidean distance with separate embeddings $\psi \not= \phi$ implies that there exists $x$ with $d(x, x) > 0$, violating identity of indiscernibles (Definiteness, M2).
>
> [1] Schaul et al., Supplementary Material to Universal Value Function Approximators (2015). Available here: http://proceedings.mlr.press/v37/schaul15-supp.pdf

---

### Official Review · AnonReviewer2 · 2019-11-04
**Official Blind Review #2**

**Rating:** 8

**Review:**

This manuscript proposes a general framework to learn non-Euclidean distances from data using neural networks. The authors provide a combination of theoretical and experimental results in support of the use of several neural architectures to learn such distances. In particular, the develop “deep norms” and “wide norms”, based either on a deep or shallow neural network. Metrics are elaborated based on norms by combining them with a learnt embedding function mapping the input space de R^n. Theoretical results are mostly application textbook results and intuitive, the overall work forms a coherent line of research bridging theory and applications that sets well justified reference approaches for this topic.

**Experience Assessment:**

I do not know much about this area.

**Review Assessment: Checking Correctness Of Derivations And Theory:**

I assessed the sensibility of the derivations and theory.

**Review Assessment: Checking Correctness Of Experiments:**

I assessed the sensibility of the experiments.

**Review Assessment: Thoroughness In Paper Reading:**

I read the paper thoroughly.

---

> ### Author Response · Authors · 2019-11-09
> **Author Response to Reviewer 2**
>
> Thank you for taking the time to carefully review our paper even though it was out of area. If you have any concerns in the remaining time, we would be happy to address them.

---

### Author Response · Authors · 2019-11-15
**Minor Revision**

We thank the reviewers for their time and comments. We have uploaded a minor revision based on the feedback. In particular, we added a paragraph to Section 4 on the relevant theoretical work as raised by Reviewer 1 and Reviewer 3. We also clarified that we use Siamese networks in Section 3, in response to a comment by Reviewer 4.

---

### Decision · Program_Chairs · 2019-12-19

**Decision:**

Accept (Poster)

**Comment:**

This paper proposes a neural network approach to approximate distances, based on a representation of norms in terms of convex homogeneous functions. The authors show universal approximation of norm-induced metrics and present applications to value-function approximation in RL and graph distance problems.

Reviewers were in general agreement that this is a solid paper, well-written and with compelling results. The AC shares this positive assessment and therefore recommends acceptance.